# Statistical Mechanics of Electrowetting

**DOI:** 10.3390/e26040276

**Published:** 2024-03-22

**Authors:** Michel Y. Louge, Yujie Wang

**Affiliations:** Sibley School of Mechanical and Aerospace Engineering, Cornell University, Ithaca, NY 14853, USA; yw2277@cornell.edu

**Keywords:** contact angle, capillarity, statistical mechanics, hysteresis, electrowetting

## Abstract

We derive the ab initio equilibrium statistical mechanics of the gas–liquid–solid contact angle on planar periodic, monodisperse, textured surfaces subject to electrowetting. To that end, we extend an earlier theory that predicts the advance or recession of the contact line amount to distinct first-order phase transitions of the filling state in the ensemble of nearby surface cavities. Upon calculating the individual capacitance of a cavity subject to the influence of its near neighbors, we show how hysteresis, which is manifested by different advancing and receding contact angles, is affected by electrowetting. The analysis reveals nine distinct regimes characterizing contact angle behavior, three of which arise only when a voltage is applied to the conductive liquid drop. As the square voltage is progressively increased, the theory elucidates how the drop occasionally undergoes regime transitions triggering jumps in the contact angle, possibly changing its hysteresis, or saturating it at a value weakly dependent on further voltage growth. To illustrate these phenomena and validate the theory, we confront its predictions with four data sets. A benefit of the theory is that it forsakes trial and error when designing textured surfaces with specific contact angle behavior.

## 1. Introduction

In a recent article, we outlined the statistical mechanics that predicts the hysteretic behavior of the equilibrium angle between a textured solid plane and a liquid drop surrounded by a gas [1]. Without resorting to any free parameter, the analysis captured the magnitude of the angle upon the advance and recession of the drop in terms of the known microscopic geometry of cavities pitting the plane and surface energies of the liquid–gas, gas–solid and solid–liquid interfaces. Recognizing ‘advance’ and ‘recession’ of the contact line as first-order phase transitions in the course of filling cavities, our treatment naturally interpreted the difference between the advancing and receding contact angle cosines as hysteresis. From this calculation, we uncovered six distinct regimes of the cavity ensemble, thereby reproducing known phenomenology of the contact line.

For example, the analysis identified two of these regimes as the non-hysteretic ‘dry’ and ‘wet’ Cassie–Baxter states [2]. Crucially, we also showed that the ‘Wenzel state’, often mentioned in the capillary literature, in fact spans the other four regimes. However, unlike Wenzel’s geometrical conjecture that cavities effectively present a larger solid surface to the drop [3], our approach managed to quantify the hysteresis that is the hallmark of this state but that Wenzel himself did not explicitly predict. The statistical mechanics then naturally interpreted the pinning of drops as an energy barrier associated with the difference between the advancing and receding contact angle cosines. It also prescribed solid textures with super-hydrophobic, super-hydrophilic, or non-equilibrium behavior. Lastly, it identified two regimes exhibiting ‘metastability’, by which extrinsic action, such as exerting an external pressure or condensing a vapor, is required to force a drop to reach its equilibrium contact angle in recession [4,5].

Our treatment differed substantially from conventional analyses of capillarity, which describe meticulously the microscopic orientation and occasional pinning of liquid near the contact line, as it moves over every nook and corner of the textured solid surface. Instead, we ignored such detail and adopted the statistical mechanics of Maxwell [6] and Boltzmann [7], who disregarded the outcome of every individual collision in a molecular gas but rather handled their collective effects on the distribution of molecular velocities. A cornerstone of Boltzmann’s insight was to recognize that molecular chaos, instead of making the outcome of molecular collisions hopelessly unpredictable, allowed him to treat their aftermath statistically. A benefit of applying Maxwell–Boltzmann’s approach to capillarity is to predict unambiguously how microscopic surface geometry affects the contact angle and other phenomena where interfacial energy is paramount, such as unsaturated porous media at equilibrium or permeated by fluid flow [8,9].

In this context, our statistical mechanics of the triple contact line superceded the conventional trial-and-error practice to designing the texture of super-hydrophobic surfaces. By forsaking adjustable parameters, its results were also subject to experimental validation. In the process, an intriguing prediction arose in comparisons with data of Onda, Shibuichi, et al. [10,11]. By varying the chemical composition of the liquid, those authors changed interfacial energies and made the textured solid progressively more hydrophilic. Doing so, they explored four of the six regimes that our theory identified. Crucially, the data suggested that the contact angle jumps sharply as regime II gives way to the more hydrophilic ‘wet’ Cassie–Baxter state that we call regime IV (Figure 11, reference [1]).

Initially, we imagined that electrowetting could confirm the existence of this jump. In the conventional view of this technique [12], establishing a voltage difference between a drop and its underlying substrate is regarded as a convenient alternative to changing surface wettability. However, to our surprise, adding electrostatic energy to our existing treatment of textured surfaces was not equivalent to merely tuning interfacial energies. Instead, the introduction of interacting capacitances on the microscopic scale unveiled much richer physics.

In this article, our objective is to illustrate with a periodic texture how statistical mechanics of the contact line may be extended to electrowetting, to elucidate peculiar phenomena that arise when electrostatic energy is turned on and, toward guiding the design of suitable validation experiments, to reveal how electric fields affect regimes of contact line behavior.

We begin with a summary of electrowetting and its conventional treatment. We then derive how microscopic capacitances contribute to the energy of individual unit cells. Then, adopting the same procedure as our original statistical mechanics of textured surfaces, we calculate the contact angle and its hysteresis in closed form while uncovering three more regimes that only exist when an electric field is applied. Lastly, to demonstrate that the theory can capture a diversity of observed behavior, we validate its results against four data sets reporting phenomenologies that no single treatment had hitherto explained.

## 2. Background

In the electrowetting technique, a grounded electrode is fixed underneath a sheet of dielectric solid [13]. A liquid drop at rest on the solid is supplied with another voltage, thereby establishing a net capacitance $\bar{C}$ across the sheet beneath the liquid–solid contact patch. If the liquid is highly conductive and does not feature an electric double layer above the patch [14], it may be considered a domain of constant voltage *U*. In this case, to maintain thermodynamic equilibrium among the liquid–gas, gas–solid and solid–liquid interfaces of the drop with respective surface energies $\gamma_{𝓁g}$, $\gamma_{gs}$ and $\gamma_{s𝓁}$, electrostatic energy must be *extracted* from the solid–liquid contact interface of area *A*.

If the sheet of dielectric constant $K_{s}$ and thickness *H* was rigorously flat, its capacitance would amount to $\bar{C}≃\upsilon_{0}K_{s}A/H$, where $\upsilon_{0}≃8.854\;\mathrm{fF}/\mathrm{mm}$ is the permittivity of free space. In that instance, the contact angle cosine would rise from the ‘Young’ equilibrium value $\cos\theta_{e}\equiv \left(\gamma_{gs}-\gamma_{s𝓁}\right)/\gamma_{𝓁g}$ to its ‘Lippmann’ counterpart $\cos\theta_{e}+\xi$ or, equivalently, goniometry would record a contact angle cosine greater by the amount
(1)$$\xi \equiv \frac{1}{2}\left(\frac{\upsilon_{0}K_{s}}{H\;\gamma_{𝓁g}}\right)U^{2},$$
unless $\cos\theta_{e}+\xi >+1$, in which case, a state of non-equilibrium would ensue.

Because $\xi \propto U^{2}⩾0$, applying a voltage difference across the dielectric solid sheet, regardless of its sign, should always make the sheet appear progressively more hydrophilic. If an alternating (AC) voltage is imposed on a perfectly conductive liquid at moderate frequencies, the appropriate value of $U^{2}$ is its mean-square [13]. A challenge to the Lippmann formulation is that it allows contact angles to decrease all the way to perfect wetting (cosθ=+1), despite evidence of a ‘saturation’ at an acute angle [15].

As this article will show, a textured dielectric sheet gives rise to a non-trivial behavior that has not yet been described by models assuming a uniform capacitance $\bar{C}$ across it. By deriving analytical expressions for the dependence of contact angle on texture, our model will also provide an analytical framework for testing the statistical mechanics treatment of the triple contact line with or without electrowetting. Finally, this article will suggest how deep texture could be designed to achieve a specific contact angle response to a tuning of the applied voltage.

As we found in the absence of electrostatics $\left(\xi =0\right)$, cavities pitting the solid surface in the neighborhood of the contact line form an ensemble that can be described by statistical thermodynamics [1,8]. In its simplest form, the analysis adopts the Ising assumption that individual cavities can either be empty or full, respectively denoted by the state variable σ=±1. Their interaction with nearest neighbors affects their own propensity to transition to the opposite filling state as the contact line ebbs or flows overhead, thereby producing a hysteresis of different contact angle cosines upon recession ($\cos\theta^{-}$) or advance ($\cos\theta^{+}$), with $\theta^{+}>\theta^{-}$.

Such collective transitions in the cavity ensemble resemble an avalanche, and manifest themselves as Haines jumps in unsaturated porous media [8]. They occur at specific probabilities χ that individual cavities have been overcome by the contact line. When surface texture geometry is known, the theory predicts values $\chi^{+}$ and $\chi^{-}$ at which the transition should, respectively, occur in advance and recession [1], in terms of geometrical aspect ratios and $\cos\theta_{e}$. If the resulting χ falls within the interval [0,1], then the corresponding transition can take place. If it does not, the transition is forbidden, and cavities must remain in their original filling state. Meanwhile, because any such phase transition involves a change in overall cavity surface energy, it releases ‘latent energy’ and is therefore classified as ‘first-order’.

Without electrostatic energy (ξ=0), $\chi^{+}$ is guaranteed $>\chi^{-}$. Consequently, in that case, because there are six ways to place $\chi^{+}>\chi^{-}$ with respect to the interval [0,1], the contact angle belongs to one of six distinct regimes [1]. In this article, we develop a similar analysis that accounts for the detailed contribution of individual cavities to the capacitance of the ensemble, and we derive new expressions for $\chi^{+}$ and $\chi^{-}$ in terms of microscopic capacitances on the cavity scale. Then, with ξ>0, we find another three regimes with $\chi^{+}<\chi^{-}$ that exhibit hysteresis behavior that is not present with ξ=0.

We illustrate next how to calculate these capacitances for a specific periodic texture. Crucially, because hysteresis arises from interactions with nearby cavities, capacitances must account for the filling state of their neighbors. We carry out the analysis in two steps [1]. First, from the standpoint of the ensemble of microscopic cavities, we derive their individual energy levels, including interfacial and electrostatic contributions. Invoking the mean-field and ergodic assumptions, we calculate the probabilities $\chi^{\pm }$ for their collective first-order phase transitions and the resulting latent energies. We then delineate the nine contact angle regimes from the positions of $\chi^{\pm }$ relative to [0,1].

Second, from the macroscopic standpoint of the liquid spherical cap, we find how elementary displacements of the contact line are met by changes in contact angle. Finally, we show that hysteresis ($\cos\theta^{-}\neq \cos\theta^{+}$) arises from two causes, one associated with the occurrence (or not) of transitions in the collective filling state, and the other related to the interfacial and electrostatic interactions of neighboring cavities.

## 3. Microscopic Capacitances

We illustrate the analysis with a planar textured surface consisting of a periodic monodisperse array of square pillars of side *b*, rising a height *h* above a flat base, separated by the distance *d*, and delimiting interconnected cavities etched in a solid of overall thickness H>h (Figure 1). For this geometry, an individual cavity belongs to a unit cell having four quarter-pillars on its periphery (dashed squares in Figure 1).

In this section, we find how much electrostatic energy is attributed to a unit cell, given the filling state of its neighbors. Unlike earlier studies adopting a uniform capacitance [13], we consider subtle details on the surface cavity scale. With an eye toward the thermodynamic analysis that follows, we design a lumped-parameter model providing the simplest closed-form expressions for capacitance and electrostatic energy levels of a unit cell.

We validate the model against numerical simulations of the Laplace equation carried out in domains encompassing solid material of dielectric constant $K_{s}$, as well as gas wherever it fills cavities (dashed lines in Figure 2, Figure 3, Figure 4 and Figure 5). These domains span a grid of 5×5 unit cells around the square ‘central’ cell of interest (Figure 1). Our goal is to infer the electrostatic energy $E_{c}=\left(1/2\right)CU^{2}$ of the central cell from the surface charge density $\nu =-K_{s}\upsilon_{0}\partial V/\partial y$ induced within its footprint of area ${\left(b+d\right)}^{2}$ below the solid from the recorded potential gradient ∂V/∂y at the base. Such charge density produces a cell capacitance $C=-K_{s}{\left(b+d\right)}^{2}\upsilon_{0}\left(\partial V/\partial y\right)/U$. In the simulations, the liquid conducts perfectly, thereby imposing a uniform voltage *U* on any of its interfaces with the solid or gas. A reference voltage of zero is applied below the solid. For simplicity, the displacement field has a vanishing component perpendicular to any gas–solid interface, and to the four lateral sides of the domain. The latter are cast far enough, and finite elements are sufficiently small, that adding more cells or refining discretization leads to no greater precision in the energy of the central cell.

Because in the Ising model a cavity is either full or empty, and bulk liquid may or may not lie overhead, there are four principal cases in this problem (Table 1). In cases 1 and 2, the central cavity is full (σ=−1). There is no liquid above the domain in case 1 (χ=0, Figure 2), whereas liquid has instead overcome all cavities in case 2 (χ=1, Figure 3). In case 3, the central cavity is empty (σ=+1), and no liquid resides overhead (χ=0, Figure 4). Therefore, its unit capacitance vanishes, unless at least one of its neighboring cells is wet. In case 4, the central cavity is also dry (σ=+1), but overhead liquid guarantees a finite capacitance (χ=1, Figure 5).

Adjacent unit cells do not necessarily hold the same filling state as the ‘central’ cavity in every case. If so, we will show that electrostatic interactions among neighbors lead to additional contact angle hysteresis beyond what theory predicts without electrowetting [1]. Our simulations indicate that such interactions are only significant when ‘edge’ cavities are filled differently than the central one. In contrast, ‘corner’ cells located across pillars have negligible effects. Next-nearest neighbors are equally insignificant.

Meanwhile, because bulk liquid resides over many unit cells, cavities in case 1 only experience nearest edge cavities in cases 1 or 3. Consequently, their electrostatic energy depends on the integer number *n* of nearest edge cells in case 3. Conversely, unit cell energy in case 3 depends on the number of adjacent cells in case 1. Similarly, case 2 energy depends on the number of neighbor cells in case 4, and case 4 energy on that number in case 2. Because there are four edge cells in this square packing, 0⩽n⩽4. (If pillars were arrayed instead on a triangular packing, there would be up to 3 edge cells and no corner cell.) The instance n=0 is homogeneous filling without any near-neighbor interaction. Crucially, our simulations also reveal that, while in general the capacitance of the central unit cell depends on *n*, the actual placement of unlike nearest neighbor edge cells is inconsequential.

In all capacitance calculations, we make lengths dimensionless with the overall thickness *H* of the textured solid, and capacitances as $C^{*}\equiv CH/K_{s}\upsilon_{0}\left(A_{0}+A_{s}\right)$, where $A_{0}$ and $A_{s}$ are, respectively, areas of cavity opening and pillar cross-section. In the example of Figure 1, $A_{0}={\left(b+d\right)}^{2}-b^{2}$ and $A_{s}=b^{2}$. We denote the resulting quantities by an asterisk^*^.

Appendix A outlines a lumped-parameter model for the capacitance of the central cell, including possible interactions with unlike nearest neighbors. Without such interactions, the cavity ensemble is homogeneous and respective capacitances in the four cases are $C_{1_{0}}^{*}$, $C_{2_{0}}^{*}$, $C_{3_{0}}^{*}=0$, and $C_{4_{0}}^{*}$. For 0<n⩽4 neighboring cavities of a different filling state, Appendix A also derives $C_{1}^{*}\left(n\right)$, $C_{2}^{*}\left(n\right)$, $C_{3}^{*}\left(n\right)$ and $C_{4}^{*}\left(n\right)$.

The lumped-parameter model and its parametric approximations are not strictly necessary if surface texture is fixed. In that case, a numerical analysis of the electric field can yield capacitances in all cases required by the theory. However, for surface design optimization, such a model can reveal how the contact angle is affected by texture geometry. In general, the dimensionless capacitances $C^{*}$ depend only on geometrical aspect ratios if their electric field exclusively penetrates the dielectric solid (e.g., Equations (Equation 34)–(Equation 46)). However, if they include a capacitive interaction with adjacent neighbors involving a field traversing an empty cavity, $K_{s}$ is no longer absorbed in the definition of $C^{*}$ and it appears explicitly in the result (e.g., Equation (Equation 49) or (Equation 52) and (Equation 53)).

As Figure 6 illustrates, the model captures well the parametric dependence of the central unit cell capacitance for any number of unlike neighbors in all four cases. In the next section, we therefore adopt its expressions. There, we calculate the electrostatic energy of a unit cell in terms of its filling state σ, that of its neighbors, and the presence or absence of liquid overhead (χ=1 or 0).

## 4. Unit Cell Energy

In this section, we outline the first step in the derivation, namely the establishment of an energy statistics of unit cells in the mesoscopic ensemble of many such cells near the contact line. Because we reported the role of interfacial energies elsewhere [1], we now emphasize the electrostatic contribution.

We begin with an account of nearest neighbors in the four cases outlined earlier. In the mean-field approximation, the filling state of neighboring cells is equal to the average value $\bar{\sigma }$ in the ensemble under consideration. Therefore, the number *n* of edge cells with a different filling state than the central one must be consistent with the probability dictated by $\bar{\sigma }$. In general, the probability P(σ=−1) to find a wet cell is a linear function of σ that vanishes at σ=+1 and is unity at σ=−1. Then, P(σ=−1)=(1−σ)/2. Similarly, the probability of a dry cell is P(σ=+1)=(1+σ)/2.

Because we found earlier that the capacitance of a central cell depends on the number of neighboring edge cells, but not on their actual position, there are 4!/[n!(4−n)!] ways to place 0⩽n⩽4 unlike edge neighbors among four such edge cells. Then, in cases 1 and 2 where a central cell is wet (σ=−1), the probability to have *n* dry edge cells is $P_{1\;\mathrm{or}\;2}\left(n\right)=4!/\left[n!\left(4-n\right)!\right]\times {\left[\left(1+\bar{\sigma }\right)/2\right]}^{n}\;{\left[\left(1-\bar{\sigma }\right)/2\right]}^{\left(4-n\right)}$, which is normalized, $\sum_{n}P_{1\;\mathrm{or}\;2}=1$. Therefore, if a unit cell is wet, its mean dimensionless capacitance is
(2)$${\bar{C}}_{1\;\mathrm{or}\;2}^{*}\left(\sigma =-1;\bar{\sigma }\right)=\sum_{n=0}^{4}\frac{4!}{n!(4-n)!}{\left(\frac{1+\bar{\sigma }}{2}\right)}^{n}{\left(\frac{1-\bar{\sigma }}{2}\right)}^{4-n}C_{1\;\mathrm{or}\;2}^{*}\left(n\right),$$
where $C_{1}^{*}\left(n\right)$ and $C_{2}^{*}\left(n\right)$ are found in Equations (Equation 44) and (Equation 46), respectively. Similarly, if it is dry (σ=+1),
(3)$${\bar{C}}_{3\;\mathrm{or}\;4}^{*}\left(\sigma =+1;\bar{\sigma }\right)=\sum_{n=0}^{4}\frac{4!}{n!(4-n)!}{\left(\frac{1-\bar{\sigma }}{2}\right)}^{n}{\left(\frac{1+\bar{\sigma }}{2}\right)}^{4-n}C_{3\;\mathrm{or}\;4}^{*}\left(n\right),$$
with $C_{3}^{*}\left(n\right)$ and $C_{4}^{*}\left(n\right)$ from Equations (Equation 48) and (Equation 51).

Meanwhile, the probability of cases 1 or 3 (no bulk liquid overhead) is (1−χ), and the probability of cases 2 or 4 is χ (bulk liquid overhead). Consequently, the expected value of a wet central unit cell capacitance is
(4)$${\bar{C}}^{*}\left(\sigma =-1;\bar{\sigma };\chi \right)=\sum_{n=0}^{4}\frac{4!}{n!(4-n)!}{\left(\frac{1+\bar{\sigma }}{2}\right)}^{n}{\left(\frac{1-\bar{\sigma }}{2}\right)}^{4-n}\left[\chi C_{2}^{*}\left(n\right)+\left(1-\chi \right)C_{1}^{*}\left(n\right)\right],$$
and the expected value of a dry cell is
(5)$${\bar{C}}^{*}\left(\sigma =+1;\bar{\sigma };\chi \right)=\sum_{n=0}^{4}\frac{4!}{n!(4-n)!}{\left(\frac{1-\bar{\sigma }}{2}\right)}^{n}{\left(\frac{1+\bar{\sigma }}{2}\right)}^{4-n}\left[\chi C_{4}^{*}\left(n\right)+\left(1-\chi \right)C_{3}^{*}\left(n\right)\right],$$

As written, these expressions are indeterminate when $\bar{\sigma }=\pm 1$. However, they converge to the finite limits
(6)$$\begin{array}{ccc} {\bar{C}}^{*}\left(\sigma =-1;\bar{\sigma }=-1;\chi \right) & = & \chi C_{2_{0}}^{*}+\left(1-\chi \right)C_{1_{0}}^{*}, \end{array}$$
(7)$$\begin{array}{ccc} {\bar{C}}^{*}\left(\sigma =-1;\bar{\sigma }=+1;\chi \right) & = & \chi C_{2_{4}}^{*}+\left(1-\chi \right)C_{1_{4}}^{*}, \end{array}$$
(8)$$\begin{array}{ccc} {\bar{C}}^{*}\left(\sigma =+1;\bar{\sigma }=-1;\chi \right) & = & \chi C_{4_{4}}^{*}+\left(1-\chi \right)C_{3_{4}}^{*}, \end{array}$$
and
(9)$$\begin{array}{ccc} {\bar{C}}^{*}\left(\sigma =+1;\bar{\sigma }=+1;\chi \right) & = & \chi C_{4_{0}}^{*}+\left(1-\chi \right)C_{3_{0}}^{*}, \end{array}$$
where $C_{i_{j}}^{*}$ is short-hand for the dimensionless capacitance $C_{i}^{*}\left(n=j\right)$ of a unit cell in state *i* with *j* neighboring edge cavities of unlike filling.

To calculate energy levels for filling states σ=±1 of the central unit cell, we now find how much energy ΔE it must receive to transition from one level to another. First, we consider filling an initially dry cavity (σ=+1→−1) of volume $v_{p}$. As we showed [1], such filling involves four contributions associated with interfacial energies. They arise from the work of pressure $W=+\kappa \;\gamma_{𝓁g}\;v_{p}$ on the interface of gas and bulk liquid with curvature κ; the surface energy $\Gamma_{p}=-A_{p}\;\gamma_{𝓁g}\cos\theta_{e}$ required to replace gas by liquid on cavity walls of area $A_{p}$; the energy $\Gamma_{o}=A_{0}\gamma_{𝓁g}\left(1-2\chi \right)$ needed to produce a gas–liquid interface as the contact line advances (χ=0→1) over the cavity of opening area $A_{0}$, or to delete it as the contact line recedes (χ=1→0); and the interfacial energy $\Gamma_{n}=\bar{\sigma }\gamma_{𝓁g}A_{n}$ between the central cell and its next-nearest ‘edge’ cells of mean-field filling state $\bar{\sigma }$, where $A_{n}$ is the combined cross-section areas linking the central cell to its adjacent neighbors. In the example of Figure 1, $v_{p}=A_{0}h$, $A_{0}={\left(b+d\right)}^{2}-b^{2}$, $A_{p}=4bh+A_{0}$ and $A_{n}=4dh$.

Adopting our earlier convention [1] to make unit cell energy *E* dimensionless, $E^{*}\equiv E/\left(A_{0}\gamma_{𝓁g}\right)$, these interfacial contributions become
(10)$$\Delta E_{\mathrm{int}}^{*}\left(\sigma =+1\to -1\right)=1-2\chi -\alpha \cos\theta_{e}+\lambda \bar{\sigma }+\kappa^{*},$$
from which the parameters $\kappa^{*}\equiv \kappa v_{p}/A_{0}$, $\alpha \equiv A_{p}/A_{0}$ and $\lambda \equiv A_{n}/A_{0}$ arise. (Unless an external pressure is applied, $\kappa^{*}$ is generally too small to matter [1].) Another useful dimensionless parameter is the area fraction $\epsilon \equiv A_{0}/\left(A_{0}+A_{s}\right)$ of the top plane that consists of cavity openings. With a perfectly flat surface, ϵ=0. For the periodic texture in Figure 1,
(11)$$\epsilon =1-{\left(1+d^{*}/b^{*}\right)}^{-2},$$
(12)$$\alpha =1+\lambda b^{*}/d^{*},$$
(13)$$\lambda =4h^{*}/\left(2b^{*}+d^{*}\right)=\left(\alpha -1\right)\left[{\left(1-\epsilon \right)}^{-1/2}-1\right],$$
and
(14)$$h^{*}=\left(\alpha -1\right)b^{*}\epsilon /\left[4\left(1-\epsilon \right)\right],$$
where distances *b*, *d* and *h* are, once again, dimensionless with *H*. To these contributions, we now *subtract* the electrostatic energy $\Delta E_{\mathrm{el}}=\left(1/2\right)U^{2}\left[\bar{C}\left(\sigma =-1;\bar{\sigma };\chi \right)-\bar{C}\left(\sigma =+1;\bar{\sigma };\chi \right)\right]$ that the cavity must spend to fill, i.e., to raise its unit cell capacitance from the dry states 4 or 3 (Equation (Equation 5)) to the wet states 1 or 2 (Equation (Equation 4)). In dimensionless form, it is
(15)$$\Delta E_{\mathrm{el}}^{*}\left(\sigma =+1\to -1\right)=\xi \left[{\bar{C}}^{*}\left(\sigma =-1;\bar{\sigma };\chi \right)-{\bar{C}}^{*}\left(\sigma =+1;\bar{\sigma };\chi \right)\right]/\epsilon .$$

Overall, to transition from dry to wet, a unit cell must receive the dimensionless energy $\Delta E^{*}=\Delta E_{\mathrm{int}}^{*}-\Delta E_{\mathrm{el}}^{*}$ from Equations (Equation 10) and (Equation 15),
(16)$$\begin{array}{ccc} \Delta E^{*}\left(\sigma =+1\to -1\right)=1 & - & 2\chi -\alpha \cos\theta_{e}+\lambda \bar{\sigma }+\kappa^{*}\\ & - & \xi [{\bar{C}}^{*}\left(\sigma =-1;\bar{\sigma };\chi \right)-{\bar{C}}^{*}\left(\sigma =+1;\bar{\sigma };\chi \right)]/\epsilon . \end{array}$$

A similar calculation yields the energy supplied to a cavity for its reverse transition (σ=−1→+1) from wet to dry. Overall,
(17)$$\begin{array}{ccc} \Delta E^{*}\left(\sigma =-1\to +1\right)=-1 & + & 2\chi +\alpha \cos\theta_{e}-\lambda \bar{\sigma }-\kappa^{*}\\ & - & \xi [{\bar{C}}^{*}\left(\sigma =+1;\bar{\sigma };\chi \right)-{\bar{C}}^{*}\left(\sigma =-1;\bar{\sigma };\chi \right)]/\epsilon . \end{array}$$

In general, as its filling state evolves from −σ to +σ, the total dimensionless energy budget of a unit cell changes by the amount
(18)$$\Delta E^{*}\left(-\sigma \to +\sigma \right)=\sigma \left(2\chi -1+\alpha \cos\theta_{e}-\lambda \bar{\sigma }-\kappa^{*}\right)-\xi \left[{\bar{C}}^{*}\left(+\sigma ;\bar{\sigma };\chi \right)-{\bar{C}}^{*}\left(-\sigma ;\bar{\sigma };\chi \right)\right]/\epsilon .$$

Fixing the ground state at σ=0 and ξ=0 without loss of generality [1], the energy of a unit cell is therefore
(19)$$E^{*}=\frac{\sigma }{2}\left(2\chi -1+\alpha \cos\theta_{e}-\lambda \bar{\sigma }-\kappa^{*}\right)-\frac{\xi }{2}\left[{\bar{C}}^{*}\left(+\sigma ;\bar{\sigma };\chi \right)-{\bar{C}}^{*}\left(-\sigma ;\bar{\sigma };\chi \right)\right]/\epsilon .$$

For the periodic texture in Figure 1, dimensionless capacitances ${\bar{C}}^{*}\left(\sigma ;\bar{\sigma };\chi ;b^{*},d^{*},h^{*}\right)$ in Equations (Equation 4)–(Equation 9) and (Equation 15)–(Equation 17) are functions of the cavity filling state σ, the mean-field $\bar{\sigma }$, the probability χ and three independent geometrical aspect ratios $b^{*}$, $d^{*}$ and $h^{*}$. To investigate geometrical limits, such as when cavities shrink toward a flat solid surface, these ratios can be conveniently expressed in terms of α, ϵ and $b^{*}$ using Equations (Equation 12)–(Equation 14), such that ${\bar{C}}^{*}={\bar{C}}^{*}\left(\sigma ;\bar{\sigma };\chi ;\alpha ,\epsilon ,b^{*}\right)$. For such periodic texture, expressions of ${\bar{C}}^{*}$ are given in Equations (Equation 6)–(Equation 9).

Among the dimensionless geometrical parameters α, ϵ, λ and $b^{*}$, the first three are independent of texture scale, and therefore they should apply regardless of actual cavity size. Without electrowetting, a solid surface with microscopic features should behave as any larger homothetic texture, as long as cavities do not approach the drop size or are too large to uphold the frozen disorder condition [1,8]. In contrast, because $b^{*}$ encodes the scale *H* of the dielectric layer, size matters as soon as a voltage is applied.

## 5. Phase Transitions

After establishing transition energies like those in Equations (Equation 16) and (Equation 17), our earlier article [1] calculated the two Ising energy levels for σ=±1 in terms of geometrical parameters and the Young contact angle, χ and $\bar{\sigma }$ [1]. We then found their respective probabilities and the resulting expected value $\langle \sigma \rangle$ for the filling state of an individual unit cell, which the ergodic assumption identified with $\bar{\sigma }$. We showed that the ensemble effectively exhibits ‘frozen disorder’, such that $\langle \sigma \rangle$ can be approximated as $-\mathrm{sign}\left(\chi -\chi^{\pm }\right)$, where $\chi^{\pm }$ are values of χ that make energy in Equation (Equation 19) vanish for $\bar{\sigma }=\pm 1$ or, equivalently, that cancel the respective right-hand sides of Equations (Equation 16) or (Equation 17). For a periodic structure with single-valued (λ,α), $\chi^{\pm }$ have separate discrete values. If either one resides within the interval [0,1], then it marks the probability χ where the entire ensemble transitions from one state to the other. If not, then such a transition is forbidden.

For a contact line advancing on an initially dry surface, χ rises from 0, while the mean filling state remains pegged at $\bar{\sigma }=+1$. Therefore, ${\bar{C}}^{*}\left(-1\right)$ and ${\bar{C}}^{*}\left(+1\right)$ are given by Equations (Equation 7) and (Equation 9), respectively. From Equation (Equation 16), σ would transition from +1 to −1 at
(20)$$\chi^{+}=\frac{1-\alpha \cos\theta_{e}+\lambda +\kappa^{*}-\xi \left(C_{1_{4}}^{*}-C_{3_{0}}^{*}\right)/\epsilon }{2-\xi \left(C_{4_{0}}^{*}-C_{2_{4}}^{*}+C_{1_{4}}^{*}-C_{3_{0}}^{*}\right)/\epsilon }.$$

Conversely, for a contact line receding on an initially wet surface, $\bar{\sigma }=-1$ in Equation (Equation 17), and ${\bar{C}}^{*}\left(+1\right)$ and ${\bar{C}}^{*}\left(-1\right)$ are found in Equations (Equation 6) and (Equation 8) such that, as χ decreases from 1, σ would transition from −1 to +1 at the probability
(21)$$\chi^{-}=\frac{1-\alpha \cos\theta_{e}-\lambda +\kappa^{*}-\xi \left(C_{1_{0}}^{*}-C_{3_{4}}^{*}\right)/\epsilon }{2-\xi \left(C_{4_{4}}^{*}-C_{2_{0}}^{*}+C_{1_{0}}^{*}-C_{3_{4}}^{*}\right)/\epsilon }.$$

Because dimensionless capacitances appearing in Equations (Equation 20) and (Equation 21) can be expressed in terms of α, ϵ and $b^{*}$, the transition probabilities $\chi^{-}$ and $\chi^{+}$ are functions of α, ϵ, $b^{*}$, $\kappa^{*}$, $\cos\theta_{e}$ and ξ. In the limit where ϵ→0, these expressions appear singular; however, as Appendix A shows, they converge to finite values; nonetheless, this is inconsequential, since a rigorously flat surface experiences neither phase transition nor hysteresis.

The positions of $\chi^{\pm }$ relative to the interval [0,1] give rise to a phase diagram like that in Figure 7. This diagram is subject to geometrical restrictions. They include $h^{*}\equiv h/H<1$, since a cavity cannot be deeper than the underlying substrate; α>1, since a cavity cannot have a surface area smaller than its opening; and, from Equation (Equation 14), $b^{*}<4\left(1-\epsilon \right)/\left[\epsilon \left(\alpha -1\right)\right]$.

As Figure 7 shows, electrowetting produces richer physics than without it. In its absence, $\chi^{+}$ and $\chi^{-}$ depend linearly on $\alpha \cos\theta_{e}$, λ and $\kappa^{*}$; because $\xi =0\Rightarrow \chi^{-}<\chi^{+}$, there are six ways to place these two numbers with respect to [0,1], from which six regimes of the contact angle arise [1].

The picture is more complicated with ξ>0. Rather than straight lines separating regimes and a symmetry between hydrophilic ($\cos\theta_{e}>0$) and hydrophobic ($\cos\theta_{e}<0$) regions, boundaries are now curved. As expected, they are shifted toward the hydrophobic side of the phase diagram, thereby narrowing the range of parameters producing hydrophobic behavior. For example, the ‘dry’ Cassie–Baxter state (regime VI) covers a smaller region. Its ‘wet’ counterpart (regime IV), which was strictly confined to $\cos\theta_{e}>0$ with ξ=0, now exists with negative $\cos\theta_{e}$.

As Equations (Equation 20) and (Equation 21) indicate, transition probabilities are hysteretic if they arise, in that their value depends on the path of the contact line. Without applying a voltage (χ=0), $\chi^{-}<\chi^{+}$, and their difference is due to λ. With ξ>0, additional hysteresis arises from mutual capacitances. Curiously, $\chi^{+}$ may also become smaller than $\chi^{-}$ in rare instances, e.g., the grey regions of Figure 7 within regimes I, IV and VI. In addition, it allows three new regimes VI, VIII and IX that only exist for ξ>0.

## 6. Latent Energy

If they take place, phase transitions at $\chi^{+}$ or $\chi^{-}$ involve a latent energy, which is defined as the energy that a unit cell must receive on average to execute them. Equivalently, in a form made dimensionless with $\left(A_{0}\gamma_{𝓁g}\right)$, it is the change $L=\Delta \langle E^{*}\rangle$ across the transition in the expected value of the unit cell energy in Equation (Equation 19), where $\sigma =\bar{\sigma }$ both change sign simultaneously. To evaluate L, our earlier article first reported the expected value consistent with a filling probability that is characteristic of frozen disorder,
(22)$$P\left(\sigma =\pm 1\right)=H\left(\pm \chi_{c}\mp \chi \right),$$
where H is the Heaviside step function and $\chi_{c}=\chi^{\pm }$ is the probability at the advancing (+) or receding (−) transitions. In the Ising formulation, any function f(σ) has an expected value $\langle f\rangle =f\left(\sigma =+1\right)\times P\left(+1\right)+f\left(\sigma =-1\right)\times P\left(-1\right)$, where P(±1) is given by Equation (Equation 22). In particular, the expected capacitance terms in Equation (Equation 19) are
(23)$$\langle {\bar{C}}^{*}\left(\pm \sigma ;\bar{\sigma };\chi \right)\rangle ={\bar{C}}^{*}\left(\pm 1;\bar{\sigma };\chi \right)\times P\left(+1\right)+{\bar{C}}^{*}\left(\mp 1;\bar{\sigma };\chi \right)\times P\left(-1\right).$$

For example, at $\chi ≲\chi^{+}$ just before an advancing transition where $\sigma =\bar{\sigma }=+1$, P(+1)=1, $\langle {\bar{C}}^{*}\left(\sigma ;\bar{\sigma };\chi \right)-{\bar{C}}^{*}\left(-\sigma ;\bar{\sigma };\chi \right)\rangle ={\bar{C}}^{*}\left(+1;+1;\chi \right)-{\bar{C}}^{*}\left(-1;+1;\chi \right)$.

Without electrowetting (ξ=0), we found that advancing and receding transitions always exhibit the same ‘exothermic’ latent energy $L^{\pm }=-\lambda <0$ for a periodic textured surface. Instead, using Equation (Equation 23) to find the expected value of $E^{*}$ from Equation (Equation 19), and substituting $\chi^{\pm }$ from Equations (Equation 20) and (Equation 21), we now find that $L^{+}$ and $L^{-}$ are no longer equal, invariant or always exothermic,
(24)$$\begin{array}{ccc} L^{\pm }= & \mp & \left(2\chi^{\pm }-1+\alpha \cos\theta_{e}-\kappa^{*}\right)\\ & \mp & \frac{\xi }{2}[{\bar{C}}^{*}\left(-1;-1;\chi^{\pm }\right)-{\bar{C}}^{*}\left(+1;-1;\chi^{\pm }\right)\\ & - & {\bar{C}}^{*}\left(+1;+1;\chi^{\pm }\right)+{\bar{C}}^{*}\left(-1;+1;\chi^{\pm }\right)]/\epsilon . \end{array}$$

With ξ>0, these expressions for $L^{\pm }$ are too complicated to quote, although they are found in the Matlab program ElectroWetting.m available as Appendix A. One can calculate them from Equation (Equation 24) in terms of any of three independent geometrical parameters using Equations (Equation 6)–(Equation 9), (Equation 11)–(Equation 14), (Equation 20) and (Equation 21), and capacitances provided in Appendix A. Figure 7 illustrates their dependence on $\alpha \cos\theta_{e}$ and λ at fixed $h^{*}$. In that example, both latent energies are exothermic where they exist, $L^{\pm }<0$. However, other hydrophobic conditions can yield endothermic latent energies: for example (λ=0.2, α=8, $\theta_{e}=2\pi /3$, $h^{*}=0.9$, $K_{s}=3.2$, κ=0) has $L^{+}≃0.146$ and $L^{-}≃0.127$ for ξ=0.35 in regime I; these values also lead to the rare instance where $\chi^{+}≃0.34<\chi^{-}≃0.51$.

## 7. Macroscopic Energy Balance

To predict the advancing and receding contact angles, we now consider the evolution of the macroscopic energy of the liquid drop as the contact line travels across the underlying solid substrate pitted with cavities that are either filled or empty. As Tadmor [16] showed, elementary changes in interfacial areas contribute to a change in the potential energy of the drop
(25)$$dG=\gamma_{k𝓁}dA_{k𝓁}+\gamma_{𝓁g}dA_{𝓁g}+\gamma_{gk}dA_{gk},$$
where index *k* represents any of the three possible states of matter in unit cells beneath the line, namely gas (*g*), solid (*s*) or liquid (*ℓ*) under the Ising assumption. At the contact line, the interface area between state *k* and the liquid grows at the expense of that between gas and state *k*, so that $dA_{k𝓁}=-dA_{gk}$. Meanwhile, at constant volume, a spherical liquid cap satisfies
(26)$$dA_{𝓁g}/dA_{k𝓁}=\cos\theta ,$$
where θ is the instantaneous contact angle. Then,
(27)$$dG=\left(\gamma_{k𝓁}+\gamma_{𝓁g}\cos\theta -\gamma_{gk}\right)dA_{k𝓁},$$

For an exclusively flat solid substrate solid (k=s), Equation (Equation 27) is $dG=\left(\cos\theta -\cos\theta_{e}\right)\gamma_{𝓁g}dA_{s𝓁}$, thereby yielding how the potential energy of a drop changes as θ deviates from the equilibrium contact angle $\theta_{e}$ [1]. With an electrostatic field, Maxwell stresses induce local deformations around the contact line [17]. However, because Equation (Equation 26) remains valid even when the liquid is severely deformed by gravity [18], we conjecture that it also applies to electrowetting.

When $k=\bar{s}$ represents effective properties of the textured solid surface with cavities of known filling statistics, the corresponding elementary change in the ensemble-averaged potential energy of the drop can similarly be written
(28)$$d\bar{G}=\left(\cos\theta -\cos\theta_{\varpi }\right)\gamma_{𝓁g}dA_{\bar{s}𝓁},$$
where $dA_{\bar{s}𝓁}$ is the increment of visible contact patch area, and $\theta_{\varpi }$ represents the angle of an advancing contact line ($\theta^{+}$, ϖ=+1), or a receding one ($\theta^{-}$, ϖ=−1). In this section, we calculate these angles by evaluating the expected change $\langle \delta G\rangle$ in the drop’s potential energy as the contact line sweeps an individual unit cell, and by identifying the result with Equation (Equation 28). Equation (Equation 28) is also the basis for calculating the energy barrier hampering reversal of the contact line direction [1].

Algebraically, $\langle \delta G\rangle$ includes three ‘reversible’ contributions to drop potential that have equal and opposite expressions upon advance and recession and, if a phase transition is allowed, a possible ‘irreversible’ fourth that arises with latent energy but does not merely change sign upon contact line reversal. For instance, with ξ=0, this fourth contribution remains the same in both directions, and therefore, it produces different expressions for $\cos\theta^{+}$ and $\cos\theta^{-}$ [1].

In our earlier article, we calculated the first two average contributions $\langle \delta G_{s}\rangle$ and $\langle \delta G_{o}\rangle$, respectively attributed to the passage of the contact line over flat parts of the solid surface and over the cavity opening. Making k=s and $dA_{k𝓁}≐\varpi A_{s}$ in Equation (Equation 27),
(29)$$\langle \delta G_{s}\rangle =\varpi A_{s}\gamma_{𝓁g}\left(\cos\theta -\cos\theta_{e}\right).$$

To determine $\langle \delta G_{o}\rangle$, one considers the two Ising cases with incremental opening area $dA_{k𝓁}≐\varpi A_{0}$. If the cavity is filled, Equation (Equation 27) yields $\delta G_{o}=\varpi A_{0}\gamma_{𝓁g}\left(\cos\theta -1\right)$ with probability $P\left(-1\right)=\left(1-\bar{\sigma }\right)/2$; if it is empty, $\delta G_{o}=\varpi A_{0}\gamma_{𝓁g}\left(\cos\theta +1\right)$ with $P\left(+1\right)=\left(1+\bar{\sigma }\right)/2$. Overall, the expected value is $\langle \delta G_{o}\rangle =\varpi A_{0}\gamma_{𝓁g}\left(\cos\theta +\bar{\sigma }\right)$. Then, because $\bar{\sigma }$ depends on χ, passage of the contact line over the unit cell ensemble (χ=0→1) produces the overall integral average
(30)$$\langle \delta G_{o}\rangle =\varpi A_{0}\gamma_{𝓁g}\left(\cos\theta +\int_{\chi =0}^{1}\bar{\sigma }d\chi \right).$$

With electrowetting, a third algebraically reversible contribution arises as the drop must *give up* electrostatic energy to accommodate changes in its underlying capacitance. Like $\delta G_{o}$, elementary contributions to $\delta G_{c}$ depend on the filling state of cavities. If they are full, passage of the line brings unit cells from case 1 to case 2 without unlike neighbors ($C_{1_{0}}^{*}\to C_{2_{0}}^{*}$) and, if they are empty, from case 3 to case 4 ($C_{3_{0}}^{*}\to C_{4_{0}}^{*}$). Then, $\langle \delta G_{c}\rangle =-\left(1/2\right)U^{2}\varpi \left[\left(C_{2_{0}}-C_{1_{0}}\right)\left(1-\bar{\sigma }\right)/2+\left(C_{4_{0}}-C_{3_{0}}\right)\left(1+\bar{\sigma }\right)/2\right]$. Applying a similar integral procedure as that for $\delta G_{o}$, substituting ξ, noting that $C_{3_{0}}=0$, and defining $\delta_{c}\equiv \left(C_{2_{0}}^{*}-C_{1_{0}}^{*}\right)/C_{4_{0}}^{*}$,
(31)$$\langle \delta G_{c}\rangle =-\varpi \left(A_{0}+A_{s}\right)\gamma_{𝓁g}\frac{\xi }{2}C_{4_{0}}^{*}\left(1+\delta_{c}\right)\left[1+\frac{\left(1-\delta_{c}\right)}{\left(1+\delta_{c}\right)}\int_{\chi =0}^{1}\bar{\sigma }d\chi \right].$$

As Figure 6 suggests with $\delta_{c}≃2\%$, this expression is typically dominated by capacitance $C_{4_{0}}^{*}$ in case 4. However, different capacitance designs can also bring $\delta_{c}$ near one (Appendix B).

The fourth contribution to the incremental drop potential energy is *minus* the average latent energy that cavities must receive from the drop to switch filling state, if such first-order phase transition is allowed,
(32)$$\langle \delta G_{L}\rangle =-A_{0}\gamma_{𝓁g}L^{\pm }.$$

As expected, this expression does not depend explicitly on contact line direction (ϖ=±1, advance or recession). Adding the four contributions from Equations (Equation 29)–(Equation 32), matching the result to Equation (Equation 28) with incremental unit cell area $dA_{\bar{s}𝓁}≐\varpi \left(A_{0}+A_{s}\right)$, defining the opening surface area fraction $\epsilon \equiv A_{0}/\left(A_{0}+A_{s}\right)$, and identifying ϖ=±1 with advancing ($\theta^{+}$) and receding ($\theta^{-}$) contact angles, we find
(33)$$\cos\theta^{\pm }=\left(1-\epsilon \right)\cos\theta_{e}-\epsilon \int_{0}^{1}\bar{\sigma }d\chi +\xi C_{4_{0}}^{*}\left(\frac{1+\delta_{c}}{2}\right)\left[1+\left(\frac{1-\delta_{c}}{1+\delta_{c}}\right)\int_{0}^{1}\bar{\sigma }d\chi \right]\pm \epsilon L^{\pm }.$$

In the limit of a flat solid surface without cavities (ϵ→0), $C_{1_{0}}^{*}=\epsilon \left[1+\left(\alpha -1\right)/\left(b^{*}p_{2}\right)\right]+O\left[{\left(\alpha -1\right)}^{4}\epsilon \right]\to 0$, $C_{2_{0}}^{*}=1+\epsilon^{2}b^{*}\left(\alpha -1\right)/4]+O\left[{\left(\alpha -1\right)}^{4}\epsilon^{2}\right]\to 1$, $C_{4_{0}}^{*}=1-\epsilon^{2}b^{*}\left(K_{s}-1\right)/4]+O\left[{\left(\alpha -1\right)}^{4}\epsilon^{2}\right]\to 1$, and $\delta_{c}\to 1$. Therefore, Equation (Equation 33) converges toward $\cos\theta^{\pm }=\cos\theta_{e}+\xi$ when cavities disappear, as expected.

As Table 2 summarizes, contact angle hysteresis ($\cos\theta^{+}\neq \cos\theta^{-}$) arises from Equation (Equation 33) in two ways. In the first, hysteresis is guaranteed if a phase transition occurs in at least one direction, i.e., if $\chi^{+}$ or $\chi^{-}$ (or both) belong to the interval [0,1]. Such is the case for regimes I, II, III, VII and VIII, where the latent energy term ϵL bears a different sign in advance and recession, or vanishes in one direction. In the second way, hysteresis is inevitable whenever the term $\int \bar{\sigma }d\chi$ does not integrate to the same value in both directions, but instead depends on the initial state ${\bar{\sigma }}_{i}$ of the surface (regimes I, II, V, VII, VIII and IX). Conversely, the only non-hysteretic regimes are the wet and dry Cassie–Baxter states, which we, respectively, call regimes IV and VI.

Nonetheless, Table 2 assumes that the specified initial ${\bar{\sigma }}_{i}$ is readily achievable. However, as Quéré, Lafuma and Calliès [4,5] noted, surfaces in regimes III and V do not wet without effort, and therefore, their recession may not begin with fully filled cavities. In this instance, a goniometer would not observe as much hysteresis as what Equation (Equation 33) predicts, unless special preparation is first implemented, such as condensing water [4] or pressing the liquid on the surface [5]. Quéré, Lafuma and Calliès [4,5] interpreted this dependence on initial conditions as ‘metastability’. However, whether or not electrostatic energy is involved, our theory readily identifies the origin of such behavior [1].

## 8. Phenomenology

For the periodic array of square pillars in Figure 1, predictions of transition probabilities in Equations (Equation 20) and (Equation 21), latent energies in Equation (Equation 24), and contact angles in Equation (Equation 33) are implemented in the Matlab program ElectroWetting.m provided as Appendix A, in terms of the three independent geometrical parameters ϵ, α and $b^{*}$, the Young contact angle cosine $\cos\theta_{e}$, the dimensionless drop interface pressure $\kappa^{*}$, the solid dielectric constant $K_{s}$ and the electrowetting parameter ξ.

As written, Equations (Equation 19), (Equation 24) and (Equation 33) are general, in that they do not require that the cavity network be periodic. In principle, for any well-characterized texture, it is therefore possible to carry out electrostatic simulations of all relevant states of the central cell and its near neighbors, record the resulting capacitances, and calculate regime transitions and contact angle behavior vs voltage. Because this approach needs no empirical input beyond independently measured material dielectric constant and surface energies, we qualify it as ab initio. However, while numerical simulations can be used in principle to find capacitances $C_{i_{j}}^{*}$ of any arbitrary texture in the four cases, it may not be straightforward to establish their closed-form expressions. Then, the determination of optimum dimensions of a textured surface is best achieved by constructing a lumped-parameter model with best-fit coefficients like those used in Appendix A. Such a procedure does not impugn the generality of our framework.

Thus, for the square periodic texture of Figure 1, we fitted $p_{1}≃0.929$ and $p_{2}≃0.197$ to 36 or more homogeneous simulations (n=0) in cases 1, 2 and 4 with $0.12<b^{*}<2.2$, $0.25<d^{*}<3$, $0.11<h^{*}<0.86$ and $K_{s}=3.2$, thereby matching the lumped-parameter expressions of $C_{1_{0}}^{*}$, $C_{2_{0}}^{*}$ and $C_{4_{0}}^{*}$ to within relative errors <2.6%, <1.3% and <5.8% in those respective cases. We then conducted 20 inhomogeneous simulations with $b^{*}=d^{*}=1$ and $h^{*}=1/2$ in all four cases with 0⩽n⩽4 to least-squares fit $p_{3}≃0.044$, $p_{4}≃0.065$ and $p_{5}≃0.10$ (Figure 6).

In the example of Figure 1, all unit cells have the same rectangular dimensions, and therefore have single-valued (λ,α). If instead the texture possessed a distribution F(λ,α), multiple conformations would have to be simulated to produce ab initio predictions. With these challenges in mind, we exploit the result of Equation (Equation 33) to reveal how peculiar phenomena observed in electrowetting also arise with the ideal periodic geometry that we analyzed. We begin with two data sets obtained on random textures [19,20]. Then, we consider a periodic design [21] resembling Figure 1. Finally, we illustrate how an altogether different surface topology can be analyzed [12].

### 8.1. Blake et al. [19]

Figure 8 recalls the experiments of Blake et al. [19] to illustrate how the theory can predict contact angle saturation, whereby cosθ does not increase with voltage squared until reaching perfect wetting at cosθ=+1 (and non-equilibrium beyond), but instead switches regime to limit its rise. Because these authors obtained the data for a PET film without additional texture, comparisons of our model with those data are only qualitative. However, they suggest that saturation begins as the voltage rises across the transition between regime II and IV. Because hysteretic regimes I and II have $\cos\theta^{+}<\cos\theta^{-}$, they possess a substantial energy barrier, calculated in Reference [1], which promotes contact line pinning until saturation occurs. If a goniometry experiment could keep the drop in place in regime IV despite the absence of pinning, regime IV would eventually give way to hysteretic regime V at ξ≃1.18, then to regime VI at ξ≃1.34 (inset of Figure 8). However, because regimes V and VI have out-of-equilibrium advancing angles, they cannot be observed.

### 8.2. Gupta et al. [20]

Gupta et al. [20] conducted experiments in which they quantified hysteresis on a hydrophobic solid. Once again, because the latter’s microscopic texture is unknown, comparisons with our model are merely instructive. As Figure 9 suggests, these authors observed a stable drop until the transition from regime II to IV, at which point contact angle saturation occurred. In principle, greater voltages should raise cosθ further toward perfect hydrophily in subsequent regimes IX and VI (inset). However, the absence of an energy barrier in non-hysteretic regime IV made it impractical to keep the drop steady and explore those regimes at larger $U^{2}$. On the other hand, the sudden removal of the barrier near ξ≃0.641 allowed the drop to undergo a recession that the substantial barrier in hysteretic regimes I and II had hitherto prohibited. With effective values of ϵ, α and $b^{*}$ shown in Figure 9, the model of Equation (Equation 33) captures essential features of the data, with the exception of the apparent threshold ξ≃0.06 that electrowetting had to exceed before affecting the advancing angle.

### 8.3. Herbertson et al. [21]

Herbertson et al. [21] created a patterned solid surface featuring cylindrical posts in an array of equilateral triangles. Because each of their unit cells possessed three adjacent cavities, their arrangement was topologically different from the square packing of columns with four neighbors that we modeled. In addition, their design included ridges of height h/2 joining every two posts along parallel lines, which amounted to two and a half cross-sections separating each cavity to its neighbors, thereby reducing λ. In spite of these differences, it is instructive to compare our predictions to their data, in which they raised the voltage from zero until saturation, then progressively brought it back down again (Figure 10). As these authors suggested, we estimate the underlying scale of ξ by fixing $\left(1/2\right)\upsilon_{0}K_{s}/\left(H\gamma_{𝓁g}\right)≃7\;10^{-5}\;V^{-2}$. Otherwise, we adopt values of α, λ, ϵ and $h^{*}$ calculated from their triangular lattice, including ridges and, for compatibility with our model, we prescribed an effective $b^{*}$ from Equation (Equation 14).

Figure 10 compares our predictions to data. As these authors raised ξ from zero, the textured solid conformed to regime III in 0<ξ<0.63, then regime V in 0.63<ξ<1.07, while the advancing angle adjusted to a progressively smaller equilibrium value at each new voltage. Both regimes are deemed ‘metastable’ [4,5], as they cannot spontaneously achieve complete cavity wetting upon recession (Table 2). Since they are also hysteretic, they oppose a substantial energy barrier to retracting the contact angle [1]. Consequently, because the subsequent transition to regime II required a sharp increase in θ at ξ≃1.07, the small drop, barely deformed by gravity, likely could not recede. Instead, it never transitioned to regime II and likely remained ‘frozen’ in its current state of regime V (dash–dotted line). Then, as ξ was subsequently lowered, the contact patch area and cosθ also remained nearly invariant. In this stand-down, the metastability and energy barrier of regime V prevented the system to explore its equilibrium receding angle found in Table 2. Instead, because returning values of cosθ were located between the two ‘primary curves’ of $\cos\theta^{-}$ and $\cos\theta^{+}$, they exhibited a behavior called ‘return-point’ hysteresis [8].

Figure 10 illustrates the important fact that values of cosθ predicted by the equilibrium theory cannot always be achieved in practice. In other words, changing the voltage can occasionally induce regime transitions requiring a reversal of the contact line that the energy barrier of hysteresis may forbid. If ‘metastable’ regimes III and V are involved, some transitions can similarly be forbidden, as they are predicated on an initial state that these regimes cannot spontaneously reach without applying extrinsic energy [1].

### 8.4. Krupenkin et al. [12]

In this context, we illustrate similar limitations of the equilibrium theory by comparing its predictions with the data of Krupenkin et al. [12], who created a texture consisting of very slender microscopic posts on a square lattice with relatively large pitch and, unlike solids in Figure 2, Figure 3, Figure 4 and Figure 5, a thin dielectric layer covering all features at a constant distance *w* from their exposed surface. Because this layer sets the electrostatic energy, its uniform thickness produces different capacitances and predictions than in Section 3, Section 4, Section 5, Section 6 and Section 7. Because w≪H, the resulting dimensionless square voltages $\xi \equiv \left(1/2\right)\upsilon_{0}K_{s}U^{2}/\left(w\gamma_{𝓁g}\right)$ are also much larger than those in Figure 8, Figure 9 and Figure 10. To illustrate how the theory is applied to this case, Appendix B summarizes how predictions are updated.

Contrary to what Krupenkin et al. [12] surmised from their observations of a linear relation of cosθ vs (1−ϵ), the drop began in hysteretic and ‘metastable’ regime V rather than the non-hysteretic dry Cassie–Baxter state (regime VI). As cosθ increased with ξ, the contact angle decreased, therefore causing the constant volume drop to advance. The drop then experienced a sudden transition to regime II (Figure 11), which should have resulted in a rapid reduction in cosθ or, equivalently, a higher θ leading to recession. However, the hysteretic nature of regimes V and II erected an energy barrier that pinned the contact line and prevented θ from reversing course. In addition, the transition threw the system out of equilibrium, as the cosθ<−1 outcome of Equation (Equation 33) is impossible. Shortly afterwards, the rising square voltage caused yet another transition to regime IV, once again out of equilibrium (cosθ>+1).

As Figure 11 shows, the theory, which adopts the actual textured geometry and Young contact angle that Krupenkin et al. [12] reported, predicts the rise of cosθ with ξ in regime V and the sudden transition that they observed. However, it cannot capture with certainty cosθ behind the transition, as the system deviates sharply from equilibrium. Lastly, although the theory predicts remarkably well the transition value of ξ for the relatively short pitch of b+d=1.05μm that Krupenkin et al. [12] staged (Figure 11, left), it fares less well for the sparse forest of slender posts on the much larger 4μm pitch (Figure 11, right). Because cavities were deep and wide at 4μm, they were unlikely to possess only states of complete filling or complete drainage, thus challenging the basic assumption of the Ising theory.

## 9. Conclusions

We extended the statistical mechanics of the triple contact line to predict electrowetting behavior on a textured solid surface at equilibrium. Having identified the controlling variable ξ making the square applied voltage dimensionless and calculated interacting capacitances of microscopic cavities in the texture, we invoked the same mean-field approximation, ergodic assumption, frozen disorder and Ising state variables of the original theory [1]. We then showed that electrowetting operates in nine distinct regimes. Depending on whether first-order transitions are allowed among them, these regimes may or may not release latent surface energy and possess a hysteretic difference $\left(\cos\theta^{-}-\cos\theta^{+}\right)$ between the cosines of their receding and advancing contact angles.

Because this difference determines the magnitude of the energy barrier controlling the reversal of contact line direction, the theory implies that equilibrium regimes IV and VI, sometimes called the ‘wet’ and ‘dry’ Cassie–Baxter states, may challenge goniometry, as they allow drops to move freely without hysteretic pinning. As in the original theory, we found that the traditional ‘Wenzel state’ does not exist as such, but that it spans instead the other seven hysteretic regimes. Meanwhile, we recognized that some regimes may not materialize if they operate out of equilibrium, or if their onset requires the drop to reverse direction while the contact line is pinned. In addition, we noted that ‘metastable’ regimes III and V cannot achieve the filled initial state that they require to recede [4,5]. Lastly, contrary to a traditional conjecture, our calculations implied that raising ξ is not equivalent to changing the Young contact angle cosine of the underlying pure uniform solid surface.

To validate the theory and lend new insight to the rich physics of electrowetting, we confronted our predictions with four published sets of data. Because the solid surface in the first two sets featured uncharacterized random texture, our comparisons were merely qualitative, but they explained why cosθ ‘saturates’ as the dimensionless square voltage ξ keeps rising [19,20], and they captured the hysteretic behavior of $\cos\theta^{\pm }$ vs ξ [20].

The other two sets involved textured surfaces with known geometry and surface energies, which we adopted to test our predictions quantitatively. A common feature of hysteretic systems is that their current state depends on prior history. In this context, our theory predicts equilibrium states whether they can be achieved or not. The data of Herbertson et al. [21] illustrated this point by examining the peculiar operation of metastable regime V, which ‘froze’ cosθ at the value reached with the largest imposed ξ.

Lastly, the set of Krupenkin et al. [12], which they acquired with a forest of slender microscopic cylindrical posts, revealed further limitations of the equilibrium theory. Although our calculations closely predicted the value of ξ at which these authors observed a jump in cosθ, the model could not address its subsequent modest rise with ξ, which occurred in regime IV operating out of equilibrium. Predictions also fared less well with their sparsest forest, which challenged our Ising assumption of only two states (full or empty) for the resulting cavities. For such scant texture, definitive predictions may require detailed numerical simulations of capillary interactions at the microscopic scale, which our equilibrium statistical mechanics had conveniently allowed us to ignore.

In spite of these limitations, a benefit of the theory is that it provides definitive predictions of textured surfaces and avoids a trial-and-error approach to their design. By identifying the cause of contact angle hysteresis, pinning, saturation, metastability, regime jumps, super-hydrophobicity, super-hydrophilicity and out-of-equilibrium behavior, this statistical mechanics of the contact line, whether or not electrowetting is involved, affords insights that the traditional Wenzel theory could not provide.

## Figures and Tables

**Figure 1 entropy-26-00276-f001:**
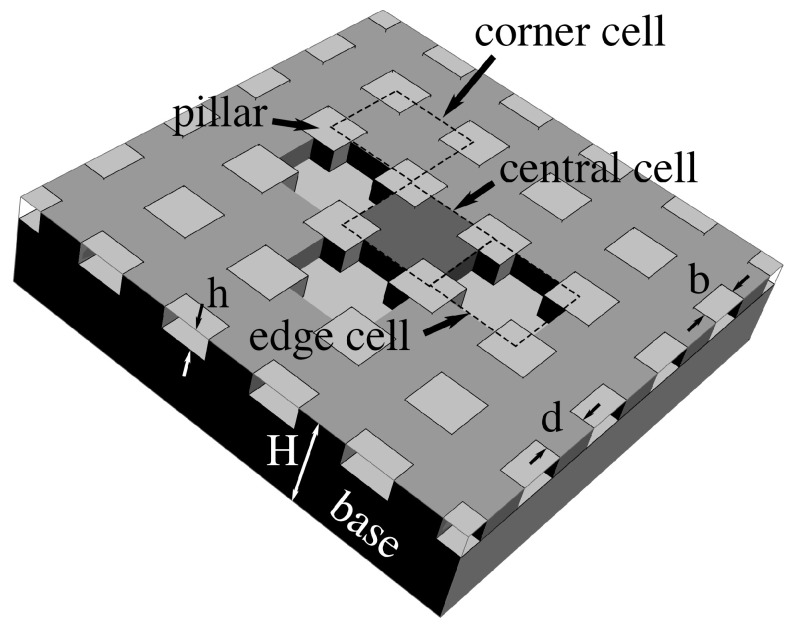
A typical subset of the cavity ensemble used in numerical simulations. In this square monodisperse periodic geometry with 5×5 unit cells, a ‘central’ cell is surrounded by four neighboring ‘edge’ cells connected through openings between pillars of square cross-section (lightly shaded), and four ‘corner’ cells across the latter. In this example, drawn for ‘case 1’ with n=3, the central cavity is filled (liquid darkly shaded), three edge cavities are empty and the rest are full of liquid (shown as another shade of gray). For simplicity, we ignore curvature of any gas–liquid interface between the central and edge cells [1,8].

**Figure 2 entropy-26-00276-f002:**
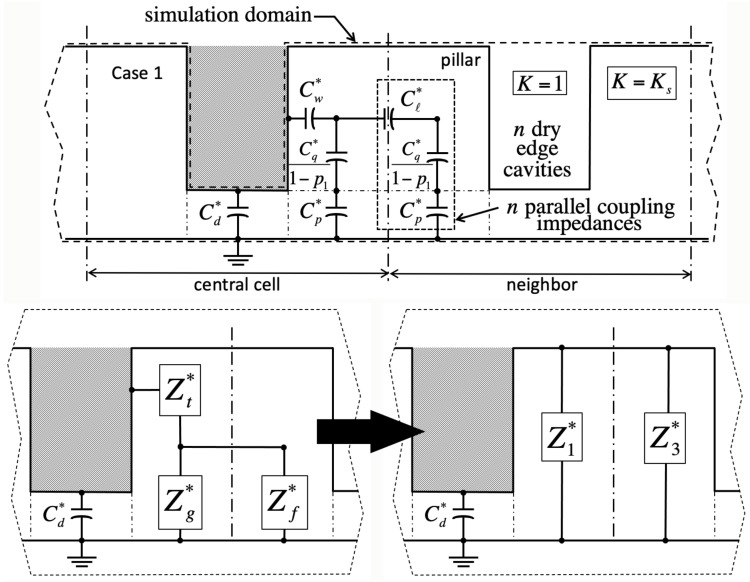
Lumped-parameter capacitance model in case 1 (filled central cavity, σ=−1; no bulk fluid overhead, χ=0). Hashed areas mark conductive liquid. **Top**: The closed dashed line delimits the numerical simulation domain, which includes solid of dielectric constant $K=K_{s}$ and gas with K=1. Thick vertical dashed–dotted lines define boundaries of unit cells; thin ones outline solid material contributing to the capacitances shown. Dry neighboring ‘edge’ cavities may produce *n* parallel capacitances combining those enclosed in the dashed rectangle. In homogeneous case 1 with n=0, the neighbor cavity would also contain liquid (not shown here). **Bottom**: Successive transformations to calculate the equivalent impedance $Z_{1}^{*}$ of the four quarter-pillars belonging to the central cell when n≠0. Because $Z_{3}^{*}$ belongs to the adjacent unit cell, it does not feature in calculations of the central cell energy. For notations, see Appendix A.

**Figure 3 entropy-26-00276-f003:**
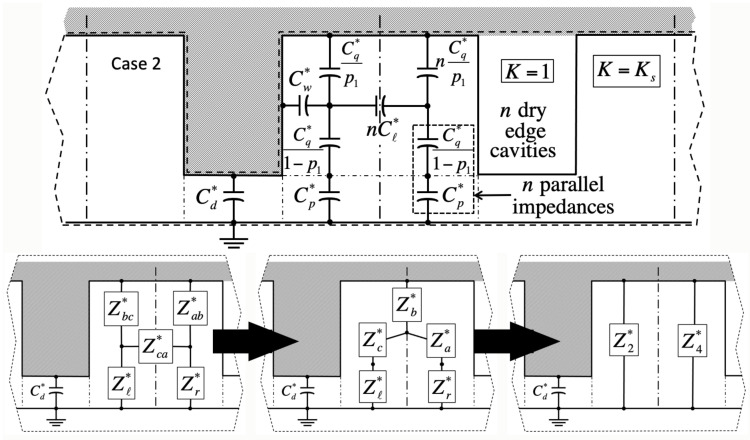
Lumped-parameter capacitance model in case 2 (filled central cavity, σ=−1; bulk fluid overhead, χ=1). **Top**: Equivalent capacitance network; lines and symbols, see Figure 2. **Bottom**: Similar to Figure 2, successive transformations to find $Z_{2}^{*}$ when n≠0. Because $Z_{4}^{*}$ belongs to the adjacent unit cell, it does not feature in calculations of the central cell energy.

**Figure 4 entropy-26-00276-f004:**
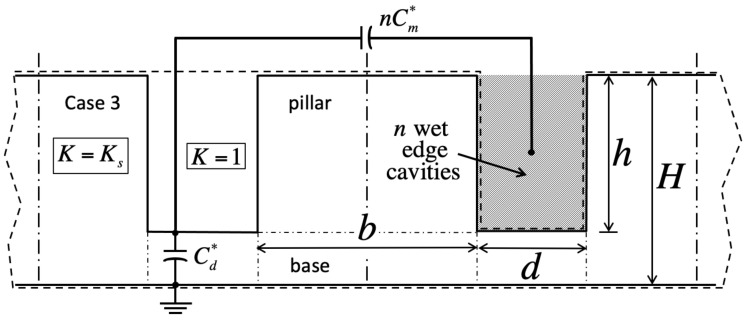
Lumped-parameter capacitance model in case 3 (empty central cavity, σ=+1; no bulk fluid overhead, χ=0). The dry central cavity is coupled to a wet neighbor through the gas–liquid interface in between pillars (capacitance $C_{m}^{*}$).

**Figure 5 entropy-26-00276-f005:**
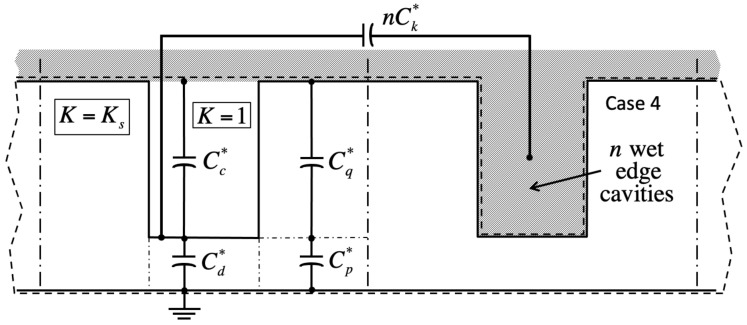
Lumped-parameter capacitance model in case 4 (empty central cavity, σ=+1; bulk fluid overhead, χ=1). Like case 3, the dry central cavity is coupled to a wet neighbor through a gas–liquid interface (capacitance $C_{k}^{*}$).

**Figure 6 entropy-26-00276-f006:**
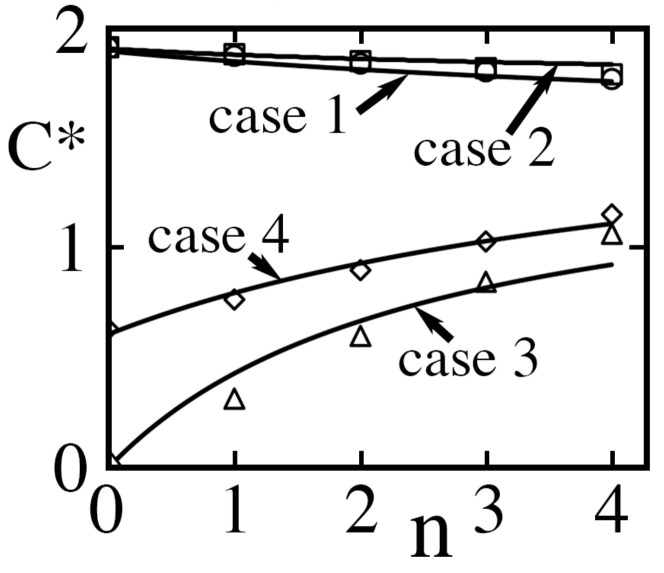
Dimensionless capacitance $C^{*}\equiv CH/K_{s}\upsilon_{0}\left(A_{0}+A_{s}\right)$ of the central unit cell vs number *n* of nearest neighbor ‘edge’ cavities of unlike state for $K_{s}=3.2$, $b^{*}\equiv b/H=1$, $d^{*}\equiv d/H=1$ and $h^{*}\equiv h/H=0.5$ or, equivalently, α=5/3, λ=2/3 and ϵ=3/4. Circles, squares, triangles and diamonds are cases 1, 2, 3 and 4, respectively. Solid lines are expressions for the corresponding lumped-parameter models of $C_{1}^{*}\left(n\right)$, $C_{2}^{*}\left(n\right)$, $C_{3}^{*}\left(n\right)$ and $C_{4}^{*}\left(n\right)$ from Equations (Equation 44), (Equation 46), (Equation 48) and (Equation 51), plotted as if *n* was a real number. The respective intercepts are $C_{1_{0}}^{*}=C_{1}^{*}\left(0\right)$, $C_{2_{0}}^{*}=C_{2}^{*}\left(0\right)$, $C_{3_{0}}^{*}=C_{3}^{*}\left(0\right)=0$ and $C_{4_{0}}^{*}=C_{4}^{*}\left(0\right)$ in Equations (Equation 39), (Equation 45), (Equation 47) and (Equation 50).

**Figure 7 entropy-26-00276-f007:**
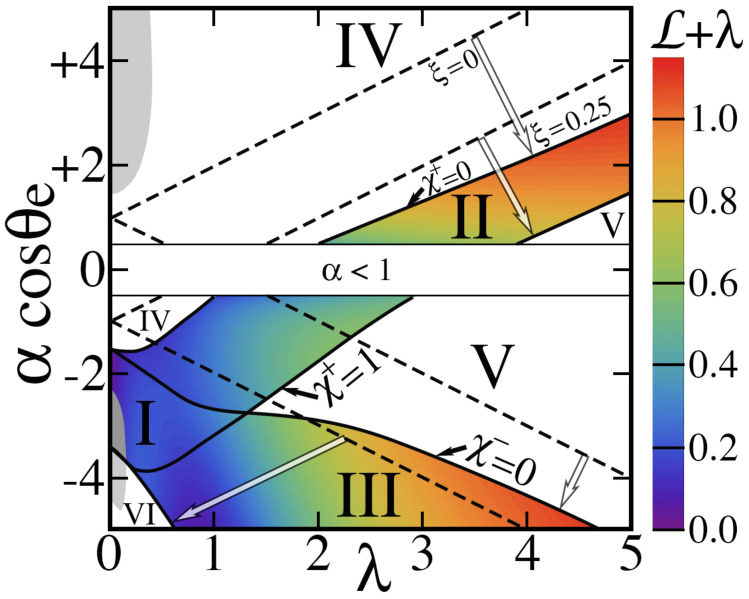
Phase diagram of $\alpha \cos\theta_{e}$ vs λ for the conditions $h^{*}=0.9$, $K_{s}=3.2$ and $\kappa^{*}=0$ with the geometry of Figure 1. **Top**: $\cos\theta_{e}=+0.5$ (hydrophilic solid). **Bottom**: $\cos\theta_{e}=-0.5$ (hydrophobic). The empty range α<1 is topologically forbidden. Solid lines are $\chi^{+}$ and $\chi^{-}=0$ or 1, as shown for ξ=0.25. Dotted lines are the corresponding straight lines for ξ=0 (no electrowetting). Light arrows point to where these lines are moved when ξ is increased from 0 to 0.25. Roman numerals mark the first six regimes. The grey regions are rare instances where $\chi^{+}<\chi^{-}$ at ξ=0.25. However, they still belong to regimes IV, I and VI where they arise. No such region exists with ξ=0. For ξ=0.25, $L^{\pm }<0$ and the color scale indicates $\lambda +L^{+}$ in regimes I and II, and $\lambda +L^{-}$ in regimes I and III. (Contrast this with $\lambda +L^{\pm }\equiv 0$ for ξ=0.) Color is not shown in regimes IV, V and VI where no transition is allowed.

**Figure 8 entropy-26-00276-f008:**
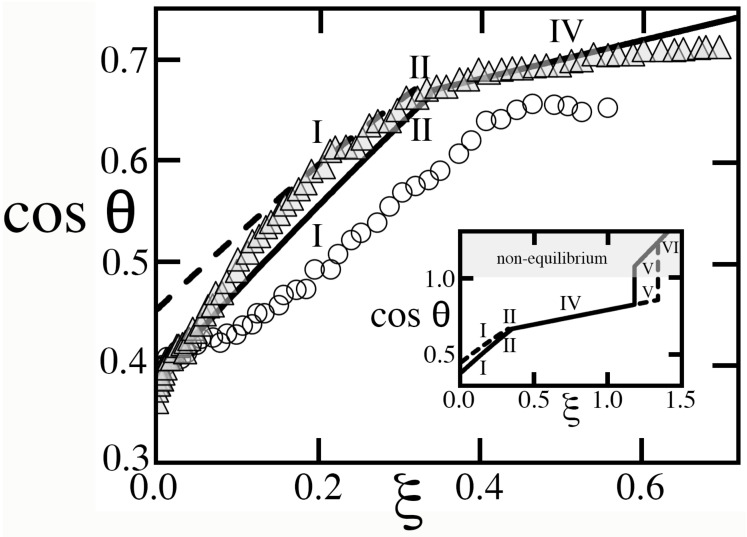
The data of Blake et al. [19] for contact angle cosines of a sessile drop with $\gamma_{𝓁g}≃0.064\;J/m^{2}$ vs ξ on a PET film with $K_{s}≃3.4$ and thickness H≃100μm. Circles and triangles are, respectively, DC and mean-square AC voltages converted to ξ using these values. (Unlike [19], we base ξ on mean-square AC voltage rather than AC amplitude [13].) Solid and dashed lines are predictions of $\cos\theta^{+}$ (advance) and $\cos\theta^{-}$ (recession) in Equation (Equation 33) with $\cos\theta_{e}≃0.39$ [15] and $\kappa^{*}=0$. Because this surface has random topography, ϵ=0.35, α=1.2 and $b^{*}=0.23$ are fitted to data only to judge whether the model can reproduce trends qualitatively. Inset: Upon increasing ξ, the advancing angle conforms to regime I in 0<ξ<0.32, then regime II (0.32<ξ<0.338), followed by a smaller saturated slope in regime IV without hysteresis (0.339<ξ<1.18), in which the absence of a pinning energy barrier makes it difficult to keep the drop from sliding away. Regime V (1.18<ξ<1.34) has inverted hysteresis, but its advancing angle is out of equilibrium. Regime VI (ξ>1.34) cannot be reached either. Regimes are shown as roman numerals.

**Figure 9 entropy-26-00276-f009:**
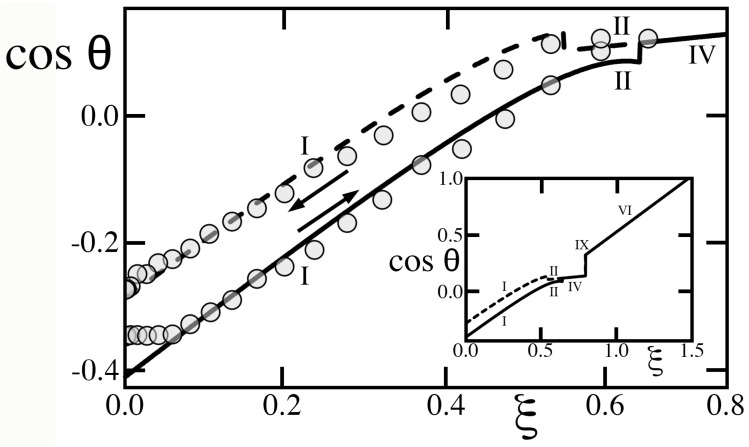
The data of Gupta et al. [20] ($\cos\theta^{+}$, bottom symbols; $\cos\theta^{-}$, top; from their Figure 2b) on a PDMS sheet (H≃10μm, $K_{s}≃2.65$) with sessile water drops ($\gamma_{𝓁g}≃0.072\;J/m^{2}$). Solid and dashed lines, see Figure 8, are for $\cos\theta_{e}=-0.29$, ϵ=0.25, α=1.8, $b^{*}=0.3$, and $\kappa^{*}=0$ with regimes I (0<ξ<0.546), II (0.546<ξ<0.641), IV (0.641<ξ<0.794), IX (0.794<ξ<0.796), VI (0.796<ξ) reaching perfect wetting at ξ≃1.48 (inset). Regimes are shown as roman numerals. Arrows mark when $U^{2}$ (and ξ) are increased or decreased.

**Figure 10 entropy-26-00276-f010:**
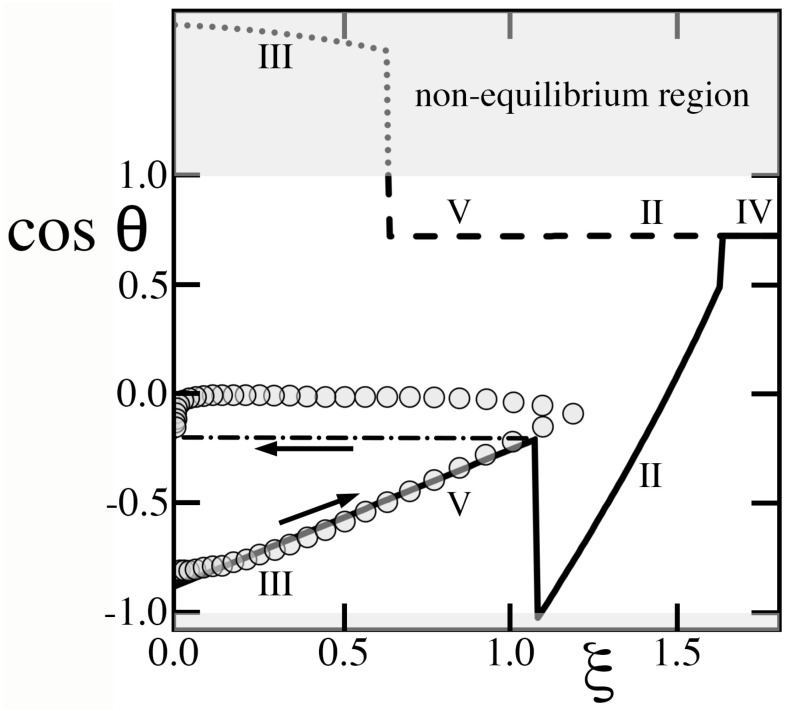
The data of Herbertson et al. [21] with posts of b=7 μm diameter, h=6.5 μm and H=15 μm on an equilateral triangular pattern of b+d=15μm center-to-center distance with a single ridge of height h/2 in each cavity, producing $\alpha =1+4h\left(\pi b+d\right)/[2\sqrt{3}{\left(b+d\right)}^{2}-\pi b^{2}]≃2.25$, $\lambda =20dh/[2\sqrt{3}{\left(b+d\right)}^{2}-\pi b^{2}]≃1.66$, $\epsilon =1-\pi {\left[b/\left(b+d\right)\right]}^{2}/2\sqrt{3}≃0.803$. For comparison, we adopt Equation (Equation 14) to find $b^{*}≃0.34$ from $h^{*}≃0.43$. Consecutive regimes as voltage is raised: Regime III (0<ξ<0.63), regime V (0.63<ξ<1.07), regime II (1.07<ξ<1.62), regime IV (1.62<ξ). Solid, dashed lines and arrows, see Figure 9. The dotted line and shaded regions mark non-equilibrium values of $\cos\theta^{-}>+1$. The dash–dotted line marks the ‘frozen’ cosθ as the drop fails to transition from regime V to regime II, which the theory underpredicts until voltage excitation is released.

**Figure 11 entropy-26-00276-f011:**
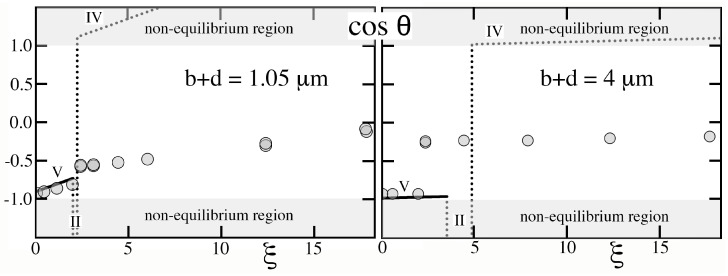
The data of Krupenkin et al. [12] with molten salt of $\gamma_{𝓁g}≃0.062\;J/m^{2}$ on cylindrical posts of b=0.35μm diameter and h=7μm height on square patterns of pitch b+d=1.05μm (**left**, α≃8.65, λ≃19.5, ϵ≃0.913) and 4μm (**right**, α≃1.48, λ≃6.43, ϵ≃0.994). In this design, the dielectric layer of $K_{s}≃3.8$ conforms everywhere to the textured surface with constant thickness w=0.054μm resulting in microscopic capacitances calculated in Appendix B. The definition of $\xi \equiv \left(1/2\right)\upsilon_{0}K_{s}U^{2}/\left(w\;\gamma_{𝓁g}\right)$ is now based on *w* rather than *H*. Similarly, the scale of texture is relative to *w*, e.g., $b^{*}≃6.36$. Krupenkin et al. [12] reported $\cos\theta_{e}≃0.136$.

**Table 1 entropy-26-00276-t001:** Four cases in a periodic rectangular array of four pillars delimiting a central cell surrounded by four edge neighboring cells and four corner cells. σ is the filling state of the central cell. χ is the probability for the presence of the liquid drop overhead.

Case	σ	χ	Figure
1	−1	0	Figure 2
2	−1	1	Figure 3
3	+1	0	Figure 4
4	+1	1	Figure 5

**Table 2 entropy-26-00276-t002:** Terms in Equation (Equation 33). ${\bar{\sigma }}_{i}$ and ${\bar{\sigma }}_{f}$ are, respectively, the initial and final filling states of the textured surface. If Equations (Equation 20) or (Equation 21) yield $0⩽\chi^{\pm }⩽1$, a phase transition is allowed with $L=L^{\pm }$ from Equation (Equation 24); otherwise, L≡0. (a) Regimes III and V can have ‘metastable’ recession, whereby an advancing contact line cannot wet cavities without intervention [1,4,5]. At its most extreme, such recession therefore begins with dry cavities and forsakes hysteresis, whereby the four columns on the right are replaced by their advancing counterparts. (b) Regime IV begins with filled cavities (${\bar{\sigma }}_{i}=-1$), irrespective of initial state, as soon as an equilibrium contact line is formed. Similarly, regime VI always begins with a dry state (${\bar{\sigma }}_{i}=+1$). As a result, regimes IV and VI are known as ‘wet’ and ‘dry’ Cassie–Baxter states [1]. Regimes VII, VIII and IX only arise with electrowetting (ξ>0) when Equations (Equation 20) and (Equation 21) yield $\chi^{+}<\chi^{-}$.

Regime			Advance (ϖ=+1)				Recession (ϖ=−1)			
	$\chi^{+}$	$\chi^{-}$	${\bar{\sigma }}_{i}$	${\bar{\sigma }}_{f}$	$\int_{0}^{1}\bar{\sigma }d\chi$	L	${\bar{\sigma }}_{i}$	${\bar{\sigma }}_{f}$	$\int_{0}^{1}\bar{\sigma }d\chi$	L
I	∈[0,1]	∈[0,1]	+1	−1	$2\chi^{+}-1$	$L^{+}$	−1	+1	$2\chi^{-}-1$	$L^{-}$
II	∈[0,1]	<0	+1	−1	$2\chi^{+}-1$	$L^{+}$	−1	−1	−1	0
III ^a^	>1	∈[0,1]	+1	+1	+1	0	−1	+1	$2\chi^{-}-1$	$L^{-}$
IV ^b^	<0	<0	−1	−1	−1	0	−1	−1	−1	0
V ^a^	>1	<0	+1	+1	+1	0	−1	−1	−1	0
VI ^b^	>1	>1	+1	+1	+1	0	+1	+1	+1	0
VII	<0	∈[0,1]	−1	−1	−1	0	−1	+1	$2\chi^{-}-1$	$L^{-}$
VIII	∈[0,1]	>1	+1	−1	$2\chi^{+}-1$	$L^{+}$	+1	+1	+1	0
IX	<0	>1	−1	−1	−1	0	+1	+1	+1	0

## Data Availability

Appendix A include the Matlab code ElectroWetting.m implementing predictions for square arrays of pillars and ElectroWettingKrupenkin.m for the topology of Krupenkin et al. [12].

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
