# Peer review of "Statistical Mechanics of Electrowetting"

_entropy, 2024, doi:10.3390/e26040276_

Round 1

Reviewer 1 Report

Comments and Suggestions for Authors

In this work, the authors derive an ab initio equilibrium statistical mechanics of the gas-liquid-solid contact angle on planar periodic, monodisperse, textured surfaces subject to electrowetting. This is an interesting work, however, the schematic of Figure 1 explicitly indicates that the approach is still based on in-situ approximations of electrowetting. I do not think this is an ab initio method.

Author Response

We thank Reviewer 1 for their service and are gratified that they deemed this article interesting.

Our use of the term ab initio implies that calculations were derived from clearly-stated assumptions and conducted without empirical artifacts. Although we used the periodic rectangular pattern of Fig. 1 to illustrate these calculations, our solution framework remains valid in any geometry. In principle, for any well-characterized texture, it is possible to carry out electrostatic simulations of all relevant states of the central cell and its near-neighbors, record the resulting capacitances, and calculate regime transitions and contact angle behavior vs voltage. In this way, no empirical fit is invoked.

Another important objective of the article was to highlight how textural aspect ratios affect regimes of the contact angle. To that end, we developed a lumped-parameter model of capacitances where these geometrical ratios could be varied while closely approximating the dependence of the energy of the central cell on the state of its neighbors. To that end, the model employed five parameters that were fitted to numerical simulations of many textures. However, this lumped-parameter capacitance model and its parameters were not essential; we only invoked them for illustration purposes; and they did not affect the general predictions of contact angle cosines in Eq. (32).

So, on balance, because qualifying our approach as free of empirical input will let readers appreciate its exactitude, we suggest that the words "ab initio" be maintained. Nonetheless, to help them distinguish the particular example from the more general framework, we added lines 454-460 and 462-465 to section 8 (colored in red for convenience), as follows:

"As written, Eqs. (19), (24) and (33) are general, in that they do not require that the cavity network be periodic. In principle, for any well-characterized texture, it is therefore possible to carry out electrostatic simulations of all relevant states of the central cell and its near-neighbors, record the resulting capacitances, and calculate regime transitions and contact angle behavior vs voltage. Because this approach needs no empirical input beyond independently-measured material dielectric constant and surface energies, we qualify it as ab initio. However, while numerical simulations can be used in principle to find capacitances C*ij of any arbitrary texture in the four cases, it may not be straightforward to establish their closed-form expressions. Then, the determination of optimum dimensions of a textured surface is best achieved by constructing a lumped-parameter model with best-fit coefficients like those in Appendix A. Such procedure does not impugn the generality of our framework."

Reviewer 2 Report

Comments and Suggestions for Authors

I enjoyed this article despite the somewhat dizzying aspects of the model. Clearly, however, this paper represents a novel, thorough, and meticulously planned consideration of electrowetting hysteresis. I only wish I had more time to dig into the specifics of the model. Having said that, I have minor comments that could improve the paper.

1. The references are mostly old except for one from 2023. Are there additional more recent applicable works that could be cited?

2. Figure 1: I believe the figure shows light-shaded pillars surrounded by liquid except where there are empty space around the pillars in the middle. The empty spaces represent air. Is my understanding correct? If not, perhaps some clarification could be added to the figure or the manuscript.

3. Would it help with readability if the cases were summarized in a table? I mention this knowing that the cases are already shown graphically in the Figures.

4. Figure 6 caption: very minor comment but I think the word “dimensionless” should go at the start of the caption before the word “capacitance”.

5. Is the equation on line 299 supposed to be a sinusoidal instead of “-sign(X-X+/-)”?

6. I love that the MATLAB code is provided in the Supplementary Material.

Author Response

I enjoyed this article despite the somewhat dizzying aspects of the model. Clearly, however, this paper represents a novel, thorough, and meticulously planned consideration of electrowetting hysteresis. I only wish I had more time to dig into the specifics of the model. Having said that, I have minor comments that could improve the paper.

The references are mostly old except for one from 2023. Are there additional more recent applicable works that could be
cited?

--- answer ---

We are grateful to Reviewer 2 for their kind words and service.

Although many recent references address ElectroWetting on Demand (EWOD) using textured solid surfaces, the literature has mainly focused on practical designs to guide small fluid drops, rather than the underlying physics of the contact angle. In this context, it was more fruitful to juxtapose our new approach with older articles that emphasized such physics.

---

Figure 1: I believe the figure shows light-shaded pillars surrounded by liquid except where there are empty space around the pillars in the middle. The empty spaces represent air. Is my understanding correct? If not, perhaps some clarification could be added to the figure or the manuscript.

--- answer ---

Reviewer 2 is correct. Heeding their advice, we clarified the first sentence in the caption of Fig. 1, as follows:

A typical subset of the cavity ensemble used in numerical simulations. In this square monodisperse periodic geometry with 5 x 5 unit cells, a `central' cell is surrounded by four neighboring `edge' cells connected through openings between pillars of square cross-section (lightly shaded), and four `corner' cells across the latter. In this example, drawn for `case 1' with n = 3, the central cavity is filled (liquid darkly shaded), three edge cavities are empty, and the rest are full of liquid (shown as another shade of gray).

---

Would it help with readability if the cases were summarized in a table? I mention this knowing that the cases are already shown graphically in the Figures.

--- answer ---

We agree that a table would help, even if it is redundant and occupies more space. Accordingly, we added a new Table 1 with the caption:

"Four cases in a periodic rectangular array of four pillars delimiting a central cell surrounded by four edge neighboring cells and four edge cells. \sigma is the filling state of the central cell. $\chi$ is the probability for the presence of the liquid drop overhead."

---

Figure 6 caption: very minor comment but I think the word 'dimensionless' should go at the start of the caption before the word 'capacitance'.

--- answer ---

We modified the caption of Fig. 6 as suggested:

Dimensionless capacitance C* = C H / K_s \upsilon_0 (A_0+A_s) of the central unit cell vs number n of nearest neighbor `edge' cavities of unlike state...

---

Is the equation on line 299 supposed to be a sinusoidal instead of '-sign(X-X+/-)' ?

--- answer ---

The sign function is correct. It is the hyperbolic tangent in the frozen disorder limit.

---

I love that the MATLAB code is provided in the Supplementary Material.

--- answer ---

We agree that it may be useful for future users of the model, whether electrowetting is applied or not.

Reviewer 3 Report

Comments and Suggestions for Authors

Review report of
Manuscript ID: entropy-2876506
Title: Statistical Mechanics of Electrowetting
Authors: Michel Y. Louge *, Yujie Wang

This paper is an extension of a wetting theory, based on statistical mechanics, published by the lead author, to electrowetting. The paper is interesting, well written and to me seems solid. The paper is probably difficult to understand without also reading the companion paper which makes it a though read (12+24 pages).
I have only a few minor remarks.

I could be helpful to include a list of (main) symbols with (a reference to) their definitions and of the scaling quantities used. Also symbols should preferentially be defined in the main text and not in an appendix. I wonder whether zeta was defined in the main text.

The data points in Fig.6 have been obtained by the numerical algorithm, but this is not mentioned explicitly. It would also be interesting to mention the values of the fitting parameters p_i used and how they have been optimized. Therefore the lines 673-678 should perhaps be moved to the main text.

line 269: I agree that the electrostatic energy must be subtracted. This may not be evident for the casual reader. Ultimately this is caused by the fact that the voltage is kept constant and the voltage source also supplies energy causing the net energy to switch sign.

Maybe it should be mentioned how the numerical data of [19][20][21][12] were obtained.
I'm puzzled by the discrepancy between the triangles and circles in Fig.8. I suppose this should correspond with Fig.7 in [19]. Here many points are almost coincident and this should also be the case in Fig.8 but this is only the case below 200 Volt.
For the experiments with no explicit periodic patterns fictitious parameters were put forward. How have these been obtained?

typo's

line 227: (4-n)! instead of (n-4)!

There seems a mix-up between the greek letter zeta (text) and xi (mainly used in the figures).

Author Response

This paper is an extension of a wetting theory, based on statistical mechanics, published by the lead author, to electrowetting. The paper is interesting, well written and to me seems solid. The paper is probably difficult to understand without also reading the companion paper which makes it a though read (12+24 pages). I have only a few minor remarks.

--- answer ---

We thank Reviewer 3 for their encouraging words and agree that the model is arduous to derive. However, we hope that, by detailing all its steps, readers will be able to extend its framework to other practical textured surface designs of their interest.

---

I could be helpful to include a list of (main) symbols with a reference to their definitions and of the scaling quantities used. Also symbols should preferentially be defined in the main text and not in an appendix. I wonder whether zeta was defined in the main text.

--- answer ---

We agree that a nomenclature would be useful. Accordingly, we produced one in the format of the journal. To avoid lengthening the core of the article, we submit it as Supplementary Material. We will leave it to the publisher to decide whether the nomenclature belongs to the main text instead.

---

The data points in Fig.6 have been obtained by the numerical algorithm, but this is not mentioned explicitly. It would also be interesting to mention the values of the fitting parameters p_i used and how they have been optimized. Therefore the lines 673-678 should perhaps be moved to the main text.

--- answer ---

We agree and moved the lines to section 8 of the main text.

---

line 269: I agree that the electrostatic energy must be subtracted. This may not be evident for the casual reader. Ultimately this is caused by the fact that the voltage is kept constant and the voltage source also supplies energy causing the net energy to switch sign.

--- answer ---

We also make this crucial point around line 85.

---

Maybe it should be mentioned how the numerical data of [19][20][21][12] were obtained. I'm puzzled by the discrepancy between the triangles and circles in Fig.8. I suppose this should correspond with Fig.7 in [19]. Here many points are almost coincident and this should also be the case in Fig.8 but this is only the case below 200 Volt. For the experiments with no explicit periodic patterns fictitious parameters were put forward. How have these been obtained?

--- answer ---

The data shown in Figs. 8-11 was obtained as discussed by their authors; we used it directly with the only exception of Blake, et al (2000) (reference [19]). As Reviewer 3 correctly noted, the Blake et al (2000) data arose from their Fig.7, which showed nearly exact coincidence of AC and DC results up to a voltage of 400V or so. However, for AC experiments, these authors "[plotted] the voltage amplitude rather than the root mean square used by other workers". We believe that such use of AC amplitude was incorrect. As Mugele and Baret (2005) articulated, [because] the liquid response depends only on the time average of the applied voltage, [... the root-mean-square must] be used". We corrected the AC data in Fig. 7 of Blake et al (2000) accordingly, thereby producing a discrepancy with the DC data. A consequence is that the DC data was not well captured by our model, although it followed a similar trend exhibiting saturation. To dispel any confusion, we added the a parenthesis to the caption of Fig. 8 "(Unlike [19], we base \xi on mean-square AC voltage rather than AC amplitude [13])."

Reviewer 3 is also correct that we invoked effective parameters in Fig. 8 (Blake et al) and Fig. 9 (Gupta, et al) to gauge whether our model could capture qualitative trends with unknown random topography, despite the model's strict derivation for periodic textured surfaces. Here, we simply varied geometrical aspect ratios of an equivalent periodic texture until the model agreed with data. Although we had already made this point clear on lines 486, 497 and 597, we added the following sentence to the caption of Fig. 8: "Because this surface has random topography, \epsilon =0.35, \alpha =1.2 and b^\ast = 0.23 are fitted to data only to judge whether the model can reproduce trends qualitatively".

---

line 227: (4-n)! instead of (n-4)!

--- answer ---

We are grateful to Reviewer 3 for having detected this typo. We have corrected it wherever it appeared.

--- 

There seems a mix-up between the greek letter zeta (text) and xi (mainly used in the figures).

--- answer ---

Unfortunately, this apparent discrepancy is entirely due to the slightly different font used by the journal for the greek letter \xi. We are at a loss how to change it, either in the text or in the glyph used in the figures. Perhaps the journal staff could help.

Round 2

Reviewer 2 Report

Comments and Suggestions for Authors

Thank you for making the requested changes. The manuscript looks good to me. One edit I would suggest is clarifying the Table 1 caption from "surrounded by four edge neighboring cells and four edge cells" to "surrounded by four edge neighboring cells and four corner cells."

Author Response

We thank Reviewer 2 for pointing out this typo, which we corrected in the revised version of the article.

Reviewer 3 Report

Comments and Suggestions for Authors

The authors did clarify/answer all my questions satisfactory except for one minor issue, which leaves me puzzled.

--- my original comment ---

Maybe it should be mentioned how the numerical data of [19][20][21][12] were obtained. I'm puzzled by the discrepancy between the triangles and circles in Fig.8. I suppose this should correspond with Fig.7 in [19]. Here many points are almost coincident and this should also be the case in Fig.8 but this is only the case below 200 Volt. For the experiments with no explicit periodic patterns fictitious parameters were put forward. How have these been obtained?

--- author's answer ---

The data shown in Figs. 8-11 was obtained as discussed by their authors; we used it directly with the only exception of Blake, et al (2000) (reference [19]). As Reviewer 3 correctly noted, the Blake et al (2000) data arose from their Fig.7, which showed nearly exact coincidence of AC and DC results up to a voltage of 400V or so. However, for AC experiments, these authors "[plotted] the voltage amplitude rather than the root mean square used by other workers". We believe that such use of AC amplitude was incorrect. As Mugele and Baret (2005) articulated, [because] the liquid response depends only on the time average of the applied voltage, [... the root-mean-square must] be used". We corrected the AC data in Fig. 7 of Blake et al (2000) accordingly, thereby producing a discrepancy with the DC data. A consequence is that the DC data was not well captured by our model, although it followed a similar trend exhibiting saturation. To dispel any confusion, we added the a parenthesis to the caption of Fig. 8 "(Unlike [19], we base \xi on mean-square AC voltage rather than AC amplitude [13])."

Reviewer 3 is also correct that we invoked effective parameters in Fig. 8 (Blake et al) and Fig. 9 (Gupta, et al) to gauge whether our model could capture qualitative trends with unknown random topography, despite the model's strict derivation for periodic textured surfaces. Here, we simply varied geometrical aspect ratios of an equivalent periodic texture until the model agreed with data. Although we had already made this point clear on lines 486, 497 and 597, we added the following sentence to the caption of Fig. 8: "Because this surface has random topography, \epsilon =0.35, \alpha =1.2 and b^\ast = 0.23 are fitted to data only to judge whether the model can reproduce trends qualitatively".

--- additional comment ---

I still don't know how the numerical data shown was obtained:

1) Did the authors have access to the original data as published in the references?

2) Did the authors derive the data from the published papers, using appropriate (scanning) software?

2) Did they do the experiments again as described in those references?

Author Response

We thank Reviewer 3 for their question. In the absence of the original data sets, we manually traced points replotted in Figs. 8 to 11 from the original articles quoted using the Matlab program below. There are several commercial and freeware alternatives.

%%%%%

function [Vdat] = TraceFigure(imStr)

% This program traces the image of a graph in .png format

% Michel Louge, MYL3@cornell.edu
% February 1, 2024
%
%    input    -    imStr   =   enter image root name as a string; 
%                               image should be in png format
%
%    output    -    Vdat    =    array; x-axis values in col 1, y-vals in col 2
%
%   global variables
%   
%           -   none    =   none
%           
% Example:
%
% Take a screenshot of the graph in .png, for example Trial.png; then type:
%
% [Vdat] = TraceFigure('Trial'); 
%
% follow prompts to trace the graph, identify axis reference pts and their
% values, indicate whether axes are linear, log-log or semi-log as prompted
%

% global none

%   internal variables
%
%           -   im      =   image array
%           -   strplot =   string type of plot
%           -   x1,x2   =   actual values of x-axis ref pts (not their log)
%           -   xpiA    =   pixel of x-axis reference points
%           -   xpix    =   x-pixel values, then rescaled
%           -   y1,y2   =   actual values of y-axis ref pts (not their log)
%           -   ypiA    =   pixel of y-axis reference points
%           -   ypix    =   y-pixel values, then rescaled
%

disp('once figure shows, click once on each point')
disp('press on return when done with all points')
disp('use first point as initial time and last point as asymptote')

%figure; % opens a new figure

eval(['im=imread(' char(39) imStr '.png' char(39) ');'])
imshow(im) % displays the image
[xpix,ypix] = ginput ; % user clicks on all points of displayed graph
%
disp('click on first then second x-axis reference points')
xpiA=ginput(2); 
%
disp('click on first then second y-axis reference points')
ypiA=ginput(2); 
%
% prompt user for values of the x- and y-axes:
%
disp('at the next prompts, enter actual values, not their log')
x1=input('input actual value of x-axis point at first click earlier >> ');
x2=input('input actual value of x-axis point at second click earlier >> ');
%
y1=input('input actual value of y-axis point at first click earlier >> ');
y2=input('input actual value of y-axis point at second click earlier >> ');
%
%
% prompt user for plot type (log-log, linear, log-linear, linear-log)
disp('at the next prompt,')
disp('enter ''linear'' for linear plot or ...')
disp('enter ''semilogx'' for log along x or ...')
disp('enter ''semilogy'' for log along y or ...')
disp('enter ''loglog'' for loglog plot')
disp...
  ('please do not forget the quotation marks '' '' , eg enter ''linear'' ')
strplot=input('now, enter the graph type as prescribed above >> ');
%
% rescale pixels:
%
if strcmp(strplot,'linear')==1 || strcmp(strplot,'semilogy')==1 % linear x
xpix = x1+(x2-x1)/(xpiA(2,1)-xpiA(1,1))*(xpix-xpiA(1,1)) ;
else
xpix = log(x1)+(log(x2)-log(x1))/...
    (xpiA(2,1)-xpiA(1,1))*(xpix-xpiA(1,1)) ; % interpolate log of x-value
xpix = exp(xpix) ; % convert logs to actual values
end
%
if strcmp(strplot,'linear')==1 || strcmp(strplot,'semilogx')==1 % linear y
ypix = y1+(y2-y1)/(ypiA(2,2)-ypiA(1,2))*(ypix-ypiA(1,2)) ;
else % y-axis has log scale
ypix = log(y1)+(log(y2)-log(y1))/...
    (ypiA(2,2)-ypiA(1,2))*(ypix-ypiA(1,2)) ; % interpolate log of y-value
ypix = exp(ypix) ; % convert logs to actual values
end

Vdat = [xpix ypix] ; % assemble the data array

%close % closes current data figure

% plot the data time-history

scrsz   = get(0,'ScreenSize')   ;   % finds screen size of current machine
figure('Position',...
 [1 scrsz(4)/2 scrsz(3)/2 scrsz(4)/2]) ; % reserves screen space for fig
                                         % 
% The ScreenSize property is a four-element vec: [left bottom width height]
% scrsz   = get(0,'ScreenSize')   ; % finds machine screen size
% figure('Name','title','Position',...
% [1 scrsz(4)/2 scrsz(3) scrsz(4)/2]) ; % reserves upper half of the screen
% get(0,'ScreenSize') % at command prompt with my big screen
% ans = [1           1        2560        1440] 

% 'Position',[left bottom width height]
% Location and size of figure's drawable area, specified as the vector, 
% [left bottom width height]. The drawable area is the inner area of the 
% window, excluding the title bar, menu bar, and tool bars. This table 
% describes each element in the Position vector:
% left =    Distance from the left edge of the primary display to the inner 
%           left edge of the figure window. This value can be negative on 
%           systems that have more than one monitor.
%
% bottom =  Distance from the bottom edge of the primary display to the 
%            inner bottom edge of the figure window. This value can be 
%           negative on systems that have more than one monitor.
%
% width  =  Distance between the right and left inner edges of the figure.
%
% height =  Distance between the top and bottom inner edges of the figure.

    if strcmp(strplot,'linear')==1          % linear plot
plot(Vdat(:,1),Vdat(:,2),'o') 
    elseif strcmp(strplot,'semilogx')==1    % semi log on x-axis
semilogx(Vdat(:,1),Vdat(:,2),'o') 
    elseif strcmp(strplot,'semilogy')==1    % semi log on y-axis
semilogy(Vdat(:,1),Vdat(:,2),'o') 
    elseif strcmp(strplot,'loglog')==1    % log-log plot
loglog(Vdat(:,1),Vdat(:,2),'o') 
    end